



# Gas-particle partitioning of polyol tracers in the western Yangtze
# River Delta, China: Absorptive or Henry's law partitioning?
Chao Qin[a], Yafeng Gou[b], Yuhang Wang[c], Yuhao Mao[b], Hong Liao[b], Qin'geng Wang[d],
Mingjie Xie[b,*]
[a] Colleges of Resources and Environmental Sciences, Nanjing Agricultural University,
Nanjing 210095, China
[b] Collaborative Innovation Center of Atmospheric Environment and Equipment
Technology, Jiangsu Key Laboratory of Atmospheric Environment Monitoring and
Pollution Control, School of Environmental Science and Engineering, Nanjing
University of Information Science & Technology, 219 Ningliu Road, Nanjing 210044,
China
[c] School of Earth and Atmospheric Sciences, Georgia Institute of Technology, Atlanta,
GA 30332
[d] State Key Laboratory of Pollution Control and Resources Reuse, School of the
Environment, Nanjing University, Nanjing 210023, China
*Corresponding to:
Mingjie Xie (mingjie.xie@nuist.edu.cn; mingjie.xie@colorado.edu);
Tel: +86-18851903788l; Fax: +86-25-58731051;
Mailing address: 219 Ningliu Road, Nanjing, Jiangsu, 210044, China



**Abstract**

Gas-particle partitioning of water-soluble organic compounds plays a significant

role in the formation and source apportionment of organic aerosols, but is poorly
characterized. In this work, gas- and particle-phase concentrations of isoprene oxidation
products (C5-alkene triols and 2-methylterols), levoglucosan, and sugar polyols were
measured simultaneously at a suburban site of the western Yangtze River Delta in east
China. All target polyols were primarily distributed into the particle phase (85.9 –
99.8%), and their average particle-phase fractions were not strictly dependent on vapor
pressures. Moreover, the measurement-based partitioning coefficients ($K_{\mathrm{p,OM}}$) of
isoprene oxidation products and levoglucosan were $10^2$ to $10^4$ times larger than their
predicted $K_{\mathrm{p,OM}}$ based on the equilibrium absorptive partitioning model. These are
likely attributed to the hygroscopic properties of polyol tracers and high aerosol liquid
water (ALW) concentrations (~20 µg m$^{-3}$) of the study location. Due to the large gaps
(up to $10^7$) between measurement-based effective Henry's law coefficients ($K_{\mathrm{H,e}}$) and
predicted values in pure water ($K_{\mathrm{H,w}}$), the gas-particle partitioning of polyol tracers
could not be depicted using Henry's law alone either. The regressions of log ($K_{\mathrm{H,w}}/K_{\mathrm{H,e}}$)
versus molality of major water-soluble components in ALW indicated that sulfate ions
("salting-in effect") and water-soluble organic carbon can promote the partitioning of
polyol tracers into the aqueous phase. These results suggest a partitioning mechanism
of enhanced aqueous-phase uptake for polyol tracers, which partly reveals the
discrepancy between observation and modeling of secondary organic aerosols.







## 1 Introduction

The water-soluble organic carbon (WSOC) documented for ambient aerosols can account for 20-80% of particulate organic matter based on carbon mass (Saxena and Hildemann, 1996; Kondo et al., 2007). Filed studies on the hygroscopic growth and cloud condensation nucleus (CCN) activity of aerosol extracts indicated that WSOC contributed significantly to aerosol hygroscopicity, and modified the hydration behavior of inorganic species (e.g., sulfate, nitrate, and ammonium; Hallar et al., 2013; Taylor et al., 2017). Thus, WSOC plays an important role in changing radiative and cloud nucleating properties of atmospheric particles. Particulate WSOC is a complex mixture of polar organic compounds containing oxygenated functional groups (e.g., hydroxyl, carboxyl, and carbonyl groups), among which a list of organic compounds with multiple hydroxyl (polyols) groups have been identified using gas chromatography (GC)-mass spectrometry (MS) and linked with specific emission sources. For example, C5-alkene triols and 2-methyltetrols are isoprene oxidation products (Claeys et al., 2004; Wang et al., 2005; Surratt et al., 2006); levoglucosan is a typical pyrolysis product of cellulose (Simoneit et al., 1999); primary saccharides (e.g., fructose and glucose) and saccharide polyols (e.g., arabitol and mannitol) are commonly associated with soil microbiota and fungal spores, respectively (Simoneit et al., 2004; Bauer et al., 2008).

To quantify the sources contributing to WSOC, concentrations of individual organic tracers are often used as inputs for receptor-based modeling (Zhang et al., 2009; Hu et al., 2010). Due to the influences of gas-particle partitioning on source apportionment, Xie et al. (2013, 2014c) suggested the involvement of gas-phase concentrations of organic makers through theoretical prediction or field measurements. The equilibrium absorptive partitioning theory outlined by Pankow (1994a, b) and



laboratory measurements of secondary organic aerosol (SOA) yields (Odum et al., 1996)
have been widely applied to predict SOA formation in traditional modeling studies
(Heald et al., 2005; Volkamer et al., 2006; Hodzic et al., 2010). The large discrepancy
between modeled and observed SOA loadings might be partly explained by the fact that
the newly generated SOA did not undergo gas-phase oxidation followed by absorptive
partitioning (Jang et al., 2002; Kroll et al., 2005; Perraud et al., 2012). Unlike non-polar
species such as $n$-alkanes and polycyclic aromatic hydrocarbons (PAHs) that are well
simulated (Simcik et al., 1998; Xie et al., 2014a), the absorptive partitioning model
underestimated particle-phase concentrations of carbonyls by several orders of
magnitude (Healy et al., 2008; Kampf et al., 2013; Shen et al., 2018;). Zhao et al. (2013)
observed a positive dependence of particle-phase pinonaldehyde on relative humidity
(RH, %), and inferred that aerosol water favored the formation of pinonaldehyde in the
atmosphere. However, every few studies have been performed on the measurement of
gaseous polyols (Xie et al., 2014b; Isaacman-VanWertz et al., 2016), and their gas-
particle partitioning were poorly understood.

Henry's law can depict the uptake of a compound into a liquid, highly dilute

solution in (e.g., cloud droplets) the atmosphere (Ip et al., 2009; Compernolle and
Müller, 2014a). Aerosol water is also a major component of atmospheric particles, and
accounts for 40% by volume at 50% RH in Europe (Tsyro, 2005). But the bulk aerosol
solution is highly concentrated with inorganic ions and WSOC. Both laboratory and
field studies observed enhanced effective Henry's law coefficients ($K^e_H$, mol m$^{-3}$ atm$^{-1}$)
of carbonyl compounds with inorganic salt concentrations (in mol kg$^{-1}$ aerosol liquid
water content, ALWC; Kampf et al., 2013; Waxman et al., 2015; Shen et al., 2018).
This termed "salting-in" effect (Setschenow, 1889) is not mechanistically understood,





and might be linked with the hydrophilic interactions (e.g., hydrogen bonding) between
polar organic compounds and inorganic ions leading to an increase of entropy or
decrease of Gibbs free energy (Almeida et al., 1983; Waxman et al., 2015). Polyol
tracers are highly water-soluble and their gas-particle partitioning is very likely driven
by the aqueous phase containing substantial ionic species in ambient aerosols. In the
Southeastern US, the particle-phase fraction of WSOC is highly dependent on RH and
ALWC (Hennigan et al., 2009).
In the present study, polyols related to specific emission sources in gaseous and
particle phases were measured concurrently in northern Nanjing, China. The sampling
and chemical analysis were performed in a similar manner as Xie et al. (2014b), while
an additional step was added prior to GC-MS analysis to clean the extracts of gaseous
samples. The absorptive and Henry's law partitioning coefficients of polyol tracers
were calculated based on measurements and predicted theoretically for comparison.
Finally, the effects of water-soluble inorganic ions and WSOC on the partitioning of
atmospheric polyols were evaluated. This work unveils the gas-particle partitioning of
polyols at a suburban site in eastern China, where the estimated average mass
concentration of aerosol liquid water is close to 20 µg m$^{-3}$ (Yang et al., 2021). The
results will benefit future studies on modeling and source apportionment of organic
aerosols.
**2 Methods**
*2.1 Field Sampling*
Details of the sampling information were provided in Yang et al. (2021). Briefly,
ambient air was sampled on the rooftop of a seven-story library building located in
Nanjing University of Information Science and Technology (NUIST 32.21 °N, 118.71




°E), a suburban site in the western Yangtze River Delta of east China. A medium
volume sampler (PM-PUF-300, Mingye Environmental, Gugangzhou, China) equipped
with a 2.5 µm cut impactor was configured to collect particulate matter with
aerodynamic diameter less than 2.5 µm ($PM_{2.5}$) and gaseous organic compounds at a
flow rate of 300 L $min^{-1}$. After the impactor, the sampled air flowed through a filter
pack containing two stacked pre-baked (550 °C, 4 h) quartz filters (20.3 cm × 12.6 cm,
Munktell Filter AB, Sweden) and a polyurethane foam (PUF, 65 mm diameter × 37.5
mm length) cartridge in series. The top quartz filter ($Q_f$) in the filter pack was loaded
with $PM_{2.5}$; gaseous organic compounds adsorbed on the backup quartz filter ($Q_b$) was
determined to evaluate sampling artifacts ("blow on" and "blow off" effects); and the
PUF cartridge was used for the sampling of gaseous polyols. Xie et al. (2014b)
demonstrated that using bare PUF material could capture gaseous 2-methytetrols and
levoglucosan with no-excessive breakthrough (< 33%). Filter and PUF samples were
collected every sixth day during daytime (8:00 AM – 7:00 PM) and night time (7:00
PM – 7:00 AM next day), respectively, from 09/28/2018 to 09/28/2019. Collection
efficiency of gaseous polyols were examined by performing breakthrough experiments
using two PUF plugs during nine sampling intervals. Prior to sampling, PUF adsorbents
were cleaned and dried in the same way as Xie et al. (2014b). Field blank filter and PUF
materials were collected every 10[th] sample for contamination adjustment. Filter and
PUF samples were sealed in prebaked aluminum foil and glass jars, respectively, at –
20 °C until analysis.
***2.2 Chemical Analysis***
**Bulk speciation.** The accumulated $PM_{2.5}$ mass and bulk components including water
soluble ions ($NH_4^+$, $SO_4^{2-}$, $NO_3^-$, $Ca^{2+}$, $Mg^{2+}$, and $K^+$), organic (OC) and elemental





carbon (EC), and WSOC were measured for each filter sample. Their final
concentrations were determined by subtracting measurement results of $Q_b$ from those
of $Q_f$. Concentrations of aerosol liquid water were predicted by ISORROPIA II model
involving ambient temperature, RH, and concentration data of $NH_4^+$, $SO_4^{2-}$, and $NO_3^-$
under the metastable state. Table S1 lists averages and ranges of ambient temperature,
RH, measured $PM_{2.5}$ components, and predicted aerosol liquid water from Yang et al.

(2021)

**Polyols analysis.** Details of the analysis method for gaseous and particulate polyols
were provided in supplementary information (Text S1). Briefly, 1/8 of each filter
sample was pre-spiked with deuterated internal standard and extracted ultrasonically
twice for 15 min in 10–15 mL of methanol and methylene chloride mixture (1:1, v/v).
After filtration, rotary evaporation, $N_2$ blown down to dryness, and reaction with 50 μL
of N, O-bis(trimethylsilyl)trifluoroacetamide (BSTFA) containing 1%
trimethylchlorosilane (TMCS) and 10 μL of pyridine, the derivatives of polyols were
diluted to 400 μL using pure hexane for GC-MS analysis. Pre-spiked PUF samples were
Soxhlet extracted using a mixture of 225 mL of methylene chloride and 25 mL of
methanol, followed by the same procedures of filter sample pretreatment. Prior to GC-
MS analysis, 50 μL of pure water was added to precipitate PUF impurities from the
final extract. As shown in Figure S1e, all PUF residues are kept in aqueous phase at the
bottom of the vial, while the derivatives of polyol tracers are supposed to be retained in
the top clear hexane solution. An aliquot of 2 μL of the supernatant was injected for
GC-MS analysis under splitless mode, and an internal standard method with a six-point
calibration curve (0.05–5 ng μL$^{-1}$) was performed to quantify polyols concentrations.
In this work, isoprene SOA products, including three C5-alkene triols (cis-2-methyl-





1,3,4-trihydroxy-1-butene,  3-methyl-2,3,4-trihydroxy-1-butene,  and  trans-2-methyl-
1,3,4-trihydroxy-1-butene; abbreviated as C5-alkene 1, 2, and 3) and two 2-
methyltetrols (2-methylthreitol and 2-methylerythritol), were quantified using meso-
erythritol; other polyols were determined using authentic standards.

Analytical recoveries of target polyols were obtained by adding known amounts of

standards to blank sampling materials (quartz filter and PUF), followed by extraction
and instrumental analysis identically as ambient samples. Method detection limits
(MDL) of individual species were estimated as three times the standard deviation of
their concentrations determined from six injections of the lowest calibration standard.
Table S2 lists recovery and MDL values of authentic standard compounds.
Concentrations of polyols in field blank samples were measured and subtracted from
air samples if necessary. To obtain appropriate gas-particle distribution of polyol tracers,
their field-blank corrected concentrations in filter and PUF samples were adjusted by
recoveries.
**Gas-particle separation and breakthrough calculation.** Polyol tracers detected in $Q_b$
samples are contributed by both gaseous adsorption and particle-phase evaporation
from $Q_f$ samples, while their relative contributions are unknown. Previous studies rarely
considered the sampling artifacts of particulate polyols. Xie et al. (2014b) adjusted
particle- and gas-phase concentrations of levoglucosan and 2-methyltetrol based on $Q_b$
measurements in two different ways. One assumed that $Q_b$ values were completely
attributed to gaseous adsorption; the other presumed equal contributions from gaseous
adsorption and $Q_f$ evaporation. However, negligible difference in gas-particle
distribution was observed. Thus particle-phase concentrations of polyols in this study
were represented by $Q_f$ values, and the gas phase was calculated as the sum of $Q_b$ and
PUF measurements.

The sampling efficiency of target polyols were evaluated by collecting and

analyzing tandemly installed PUF plugs during nine sampling intervals. The
breakthrough of each polyol was calculated as
$B = \frac{[PUF]_{back}}{[PUF]_{front}+[PUF]_{backup}} \times 100\%$ \hfill (1)
where $B$ is the breakthrough of gaseous sampling, and [PUF] represents the
concentration of specific compound in front or backup PUF sample. A value of 33%
was typically used to indicate excessive breakthrough (Peters et al., 2000).
**Calculations of partitioning coefficients**. Here, the calculations of measurement- and
theory-based absorptive partitioning coefficients ($K^m_{p,OM}$ and $K^t_{p,OM}$, $m^3$ $ug^{-1}$) were
conducted identically as those in Xie et al. (2013, 2014a, b). The equations and
parameters were detailed in supplementary information (Text S2).

As aerosol liquid water plays a significant role in the gas-particle partitioning of

water-soluble organic compounds, we proposed an equilibrium mechanism in Figure
S2. First, aerosol liquid water and water insoluble OM (WIOM) exist in two separate
phases (liquid-liquid phase separation), and WSOC and inorganic ions are totally
dissolved in the aqueous phase. The distribution of polyol tracers between aqueous and
WIOM phases is simply depicted by their octanol-water partition coefficients ($K_{OW}$)
$K_{OW} = \frac{c_{OM}}{c_w}$ \hfill (2)
where $c_{OM}$ and $c_w$ are polyols concentrations (ng $m^{-3}$ in solution) in WIOM and aqueous
phases; log $K_{OW}$ values of target polyols were given by the Estimation Programs
Interface (EPI) Suite developed by the US Environmental Protection Agency and
Syracuse Research Corporation in Table S3 (US EPA, 2012). Due to the high water-
solubility of target polyols ($K_{OW} < 0.15$), more than 95% of their particle-phase





concentrations were distributed into the aqueous phase. Second, gas-phase polyol
tracers are in equilibrium with hydrophobic OM and the aqueous phase, respectively,
following absorptive partitioning theory (eqs 1 and 2 in Text S2) and Henry's law (eq

3)

$$K_{\mathrm{H},e} = \frac{\frac{F}{M_i}}{\frac{A}{M_i} \times R \times T \times \frac{c_{\mathrm{ALW}}}{\rho_{\mathrm{w}}}} = \frac{\rho_{\mathrm{w}} \times F}{A \times R \times T \times c_{\mathrm{ALW}}} \qquad (3)$$
where $K_{\mathrm{H,e}}$ (mol m$^{-3}$ atm$^{-1}$) is the measurement-based effective Henry's law coefficient;
$F$ and $A$ (ng m$^{-3}$) represent particle- and gas-phase concentrations of polyol tracers in
ambient air; $M_i$ (g mol$^{-1}$) is the molecular weight of specific compound; $R$ (m$^3$ atm K$^{-1}$
mol$^{-1}$) and $T$ (K) are ideal gas constant and ambient temperature, respectively; $c_{\mathrm{ALW}}$ (μg
m$^{-3}$) is the mass concentration of aerosol liquid water predicted using ISORROPIA II
model; $\rho_{\mathrm{w}}$ (1 g cm$^{-3}$) is water density. For comparison purposes, the Henry's law
coefficient in pure water at 25 °C ($K^*_{\mathrm{H,w}}$) was predicted from EPI suite (Table S3), and
was adjusted for each sampling interval due to the changes in ambient temperature
using van 't Hoff equation (Text S3).
**3 Results and discussion**
***3.1 Method evaluation***

In our previous study, PUF/XAD-4 resin/PUF and PUF/XAD-7 resin/PUF

adsorbent sandwiches were tested for sampling gaseous 2-methyltetrols and
levoglucosan (Xie et al., 2014b). The results of breakthrough experiments suggested
that both the two sandwiched composites had high sampling efficiency (close to 100%).
Moreover, individual parts of the two types of composites (top PUF, middle XAD-
4/XAD-7 resin, and backup PUF) were analyzed for 7 samples, and target compounds
were only detected in top PUF. Thus, bare PUF material is suitable for sampling





gaseous 2-methyltetrols and levoglucosan.
Although PUF materials were pre-cleaned prior to sampling, a few short-chain
polyurethanes or impurities could be dissolved during Soxhlet extraction of target
compounds using the mixture of methanol and methylene chloride. These substances
precipitated when sample extracts were concentrated (Figure S1a, b), and re-dissolved
in BSTFA:TMCS/pyridine and hexane  after the derivatization step (Figure S1c, d). In
Xie et al. (2014b), an aliquot of 2 µL of the sample extract as shown in Figure S1d was
injected for GC-MS analysis. Due to the fact that the dissolved PUF materials did not
vaporize at ~300 ºC, the GC inlet liner had to be changed for cleaning every few
samples. In this work, 50 µL of pure water was added to separate PUF materials from
polyol derivatives in hexane solution. As shown in Figure S1e, all PUF residues were
retained in the aqueous solution after liquid-liquid phase separation. This pretreatment
step was added for the analysis of gaseous samples to save time for changing and
cleaning GC inlet liners. However, the revised method did not improve the recoveries
of meso-erythritol and levoglucosan in PUF samples (Table S2) compared to those in
Xie et al. (2014b). This is because the dissolved PUF materials should have an impact
on the derivatization efficiency of polyol species, and future work is warranted to
remove dissolved PUF materials in sample extracts before the derivatization step.
Measurement results of breakthrough samples and the resulting $B$ values were
shown in Figure S3. C5-alkene triols and 2-methyltetrols were mainly observed in
summertime, and levoglucosan was only detected in three pairs of breakthrough
samples. Their average $B$ values ($< 33\%$) indicated no excessive breakthrough (Figure
S3a-c), but were higher than those reported by Xie et al. (2014b). This might be ascribed
to the greater face velocity (1.5 cm s$^{-1}$) for sampling gaseous polyols than that (0.61 cm



$s^{-1}$) in our previous study. Unlike fructose which had low breakthrough (Figure S3d),
glucose and mannitol had comparable concentrations between front and backup PUF
samples for several breakthrough experiments (Figure S3e, f), indicating that PUF
materials are not suitable for sampling gaseous glucose and mannitol. Mannose and
arabitol were not detected or had BDL values for breakthrough samples, and their
breakthrough was not provided. In the current work, concentrations of polyol tracers in
filter and PUF samples were all reported, but the data of mannose, glucose, arabitol,
and mannitol in PUF samples should be treated with caution due to the lack of valid
breakthrough results.

### 278     *3.2 General description of measurement results*

Concentrations of individual polyols in $Q_f$, $Q_b$, and PUF samples are summarized
in Table S4, and their total ambient concentrations ($Q_f + Q_b + $ PUF) are depicted using
boxplots in Figure 1. Figure S4 presents temporal variations of total and $Q_f$
concentrations of individual polyols with daytime and night-time measurements
distinguished. In general, polyol tracers were predominantly observed on $Q_f$ with
averages 1-3 orders of magnitude higher than those on $Q_b$ and PUF. Levoglucosan had
the highest average total concentration ($66.1 \pm 71.1$ ng m$^{-3}$), followed by fructose (15.0
$\pm 62.9$ ng m$^{-3}$) and mannose ($14.3 \pm 31.3$ ng m$^{-3}$). C5-alkene triols and 2-methyltetrols
are formed from isoprene epoxydiols (IEPOX) under low $NO_X$ conditions (Surratt et
al., 2010). All the five species on $Q_b$ were more frequently detected and had average
concentrations 2-20 times higher than those in PUF samples. While in Xie et al. (2014b),
the sum of 2-methyltetrols in $Q_b$ and adsorbent samples were up to 2.7 times higher
than those on $Q_f$ in summer Denver, so isoprene products are not similarly distributed
between gas and aerosol phases across different regions. Moreover, isoprene-derived



polyols exhibited prominent elevations in summer (Figure S4a-e), and their daytime
concentrations ($2.02 \pm 3.73 - 10.5 \pm 29.3$ ng m$^{-3}$) were only slightly higher than those
during night-time ($1.63 \pm 4.40 - 9.65 \pm 32.7$ ng m$^{-3}$). Fu and Kawamura (2011)
investigated diurnal variations of polar organic tracers at a forest site in summer by
sampling aerosol particles every 4 h. They found that isoprene-derived SOA tracers
maximized from later afternoon to early evening. Although no IEPOX will be generated
from the oxidation of isoprene by •OH and HO$_2$• after sunset, the formations of C5-
alkene triols and 2-methyltetrols might continue until pre-existing IEPOX is exhausted.
This explains the insignificant ($p > 0.05$) day-night differences of C5-alkene triols and
2-methyltetrols in this work.

Levoglucosan was more frequently detected but far less concentrated in PUF than

in Q$_b$ samples. Its total concentrations were comparable to those in urban Denver
(average $65.3 \pm 96.8$ ng m$^{-3}$, range $2.48 - 478$ ng m$^{-3}$), where an average of ~20%
partitioned into the gas phase (Xie et al., 2014b). Due to the enhanced biomass burning
activities in cold periods for domestic heating at night, levoglucosan showed a clear
seasonal pattern (winter maxima and summer minima) and significant ($p = 0.03$) higher
concentrations during night-time (Figure S4f). Sugars and sugar alcohols are commonly
linked with soil/dust resuspension and associated microbial activities (Simoneit et al.,
2004). They were frequently detected in Q$_b$ samples with comparable averages and
ranges as those in PUF samples (Table S4). Total concentrations of fructose and glucose
were strongly ($r = 0.98$) correlated peaking in middle spring (April 2019, Figure S4h,
j), when Ca$^{2+}$ on Q$_f$ also reached its maxima of the year (Yang et al., 2021), indicating
an influence from soil/dust resuspension. Arabitol and mannitol had identical seasonal
pattern ($r = 0.89$) with elevated total concentrations from May to October (Figure S4i,


m), which might be attributed to the high levels of vegetation and autumn decomposing
(Burshtein et al., 2011). Multiple peaks of mannose concentrations were observed from
spring to autumn, suggesting a variety of contributing sources (e.g., microbial activity,
vegetation). Xylitol is likely derived from biomass burning in northern Nanjing due to
its strong correlation ($r = 0.89$) with levoglucosan.

### 3.3 Gas-particle distribution and absorptive partitioning coefficient

$Q_b$ measurements were often used to assess positive sampling artifacts of
particulate OC (Chow et al., 2010; Subramanian et al., 2004), but rarely for particle-
phase organic markers. In this study, concentrations of particulate polyols were
obtained directly from $Q_f$ measurements, and the gas phase was calculated as the sum
of $Q_b$ and PUF values. Figure S5 shows the time series of gas-phase concentrations and
particle-phase fractions ($F\%$) of individual polyol tracers. The average $F\%$ values of
measured species are linearly regressed against the logarithms of their subcooled liquid
vapor pressures at 25 °C ($p^{o,*}_L$) in Figure 2. Unlike non-polar organic tracers (e.g., $n$-
alkanes and PAHs), some polyols (e.g., 2-methyltetrols and levoglucosan) data did not
follow the linear regression line of $F\%$ versus log $p^{o,*}_L$. Although gas-phase C5-alkene
triols and 2-methyltetrols were majorly observed in summer with significant ($p < 0.05$)
day-night variations, their $F\%$ values did not show seasonality or day-night difference
($p = 0.18\text{-}0.73$). Other polyols had extremely low concentrations in the gas phase with
average $F\%$ ranging from $94.2 \pm 8.02 - 99.8 \pm 1.21\%$. The average $F\%$ values of 2-
methyltetrols ($87.5 \pm 10.6\%$) and levoglucosan ($99.8 \pm 1.21\%$) were greater than those
in urban Denver (50–80%; Xie et al., 2014b), where the average sampling temperature
($12.5 \pm 10.1$ °C) was much lower. Thus, the changes in vapor pressures with the ambient
temperature might not be the main factor driving gas-particle partitioning of polyol



tracers in northern Nanjing.

To understand if traditional absorptive partitioning theory could be applied to

predict the gas-particle partitioning of polyol tracers in northern Nanjing, Table 1
compares log $K^m_{p,OM}$ and log $K^t_{p,OM}$ of individual compounds. The average $K^m_{p,OM}$
values of isoprene SOA tracers, levolgucosan, and meso-erythritol were $10^2$ to $10^3$ times
larger than their corresponding $K^t_{p,OM}$. Comparable or even greater (up to $10^5$) gap
between $K^m_{p,OM}$ and $K^t_{p,OM}$ has been observed for carbonyls in a number of laboratory
and field studies (Healy et al., 2008; Zhao et al., 2013; Shen et al., 2018), which could
be ascribed to reactive uptake (e.g., hydration, oligomerization, and esterification) of
organic gases onto condensed phase (Galloway et al., 2009). Oligomers, sulfate and
nitrate esters of 2-methyltetrols can be formed in the aerosol phase (Surratt et al., 2010),
but these products were not expected to dominate particle-phase concentrations of 2-
methyltetrols (Lin et al., 2013; Xie et al., 2014b). Although levoglucosan can be readily
oxidized by •OH in the aqueous phase of atmospheric particles (Hennigan et al., 2010;
Hoffmann et al., 2010), the occurrence of its oligomers, sulfate or nitrate esters was not
reported in ambient aerosols. Xie et al. (2014b) found that the gas-particle partitioning
of 2-methyltetrols and levoglucosan in urban Denver were highly dependent on the
variations in ambient temperature and absorbing organic matter ($M_{OM}$). While in
southeastern US, the particle-phase fractions of isoprene SOA tracers were generally
higher than prediction based on absorptive partitioning model (Isaacman-VanWertz et
al., 2016). This discrepancy might be related to the spatial heterogeneity of ALWC,
which is expected to control the gas-particle partitioning of water-soluble organic
matter in the Eastern US (Carlton and Turpin, 2013). In this study, the large difference
between $K^m_{p,OM}$ and $K^t_{p,OM}$ indicated that some mechanisms other than absorptive



partitioning (e.g., Henry's law partitioning) should be involved to predict the gas-
particle partitioning of polyol tracers in northern Nanjing, where the ambient particles
contained substantial liquid water ($21.3 \pm 24.2$ µg m$^{-3}$; Table S1).

Unlike isoprene SOA tracers and levoglucosan, the average $K^t_{p,OM}$ values of

monosaccharides (fructose, mannose, and glucose) and sugar alcohols (xylitol, arabitol,
and mannitol) were up to $10^3$ times larger than their $K^m_{p,OM}$ (Table 1). This is probably
caused by the overestimation of gas-phase concentrations of sugar polyols. The organic
matter on $Q_b$ is mainly composed of volatile and semi-volatile organic compounds. If
the concentrations of organic compounds on $Q_b$ were comparable or higher than those
on $Q_f$, their $Q_f$ values should be dominated by positive artifact. As the vapor pressure
decreases, the evaporation loss from $Q_f$ samples becomes non-negligible. Note that the
magnitude of negative artifacts is unknown and very difficult to assess, and the vapor
pressures of monosaccharides and sugar alcohols are mostly $< 10^{-10}$ atm, their
concentrations in $Q_b$ and even PUF samples might contain more contributions from
negative artifacts than isoprene SOA tracers and levoglucosan. Considering that low-
volatile sugar polyols had less stable recoveries (Table S2) and greater breakthrough
(Figure S3e, f), caution is warranted in analyzing their $K^m_{p,OM}$ values obtained in this
study.

Figure S2 presumes that gas-phase polyols are in equilibrium with WIOM and the

aqueous phase, respectively. Then concentrations of WIOM [$1.4 \times$ (OC-WSOC)] was
used to adjust the calculation of absorptive partitioning coefficients ($K^m_{p,WIOM}$) based
on eq 1 in supplementary information. In comparison to log $K^m_{p,OM}$, the average log
$K^m_{p,WIOM}$ values of isoprene SOA tracers and levoglucosan were much closer to average
log $K^t_{p,OM}$ (Tables 1 and S5), supporting that the aerosol liquid water should have





significant impacts on gas-particle partitioning of polyol tracers.
***3.4 Effective Henry's law coefficient***
Table 2 lists the statistics of measurement-based log $K_{H,e}$ and predicted log $K_{H,w}$.
The average $K_{H,w}$ values of isoprene SOA tracers, levoglucosan, and meso-erythritols
were 2-6 orders of magnitude lower than their corresponding average $K_{H,e}$, indicating
that the ambient atmosphere in northern Nanjing favored the condensation of these
polyols. Other polyol compounds exhibited less difference between log $K_{H,e}$ and log
$K_{H,w}$, which was very likely caused by the overestimation of their gas-phase
concentrations. A number of previous studies observed enhanced $K_{H,e}$ of carbonyls with
salt concentrations in aqueous solution (Ip et al., 2009; Kampf et al., 2013; Waxman et
al., 2015; Shen et al., 2018), and described this "salting-in" effect using
$$\mathrm{Log}\left(\frac{K_{H,w}}{K_{H,e}}\right) = K_s c_{\mathrm{salt}} \qquad\qquad (4)$$
where $K_s$ is the salting constant, and $c_{\mathrm{salt}}$ is the aqueous-phase concentration of salt in
mol kg$^{-1}$ ALWC. This equation is originally defined in Setschenow (1889) by plotting
log $(K_{H,w}/K_{H,e})$ versus the total salt concentration (mol L$^{-1}$).
As sulfate has been identified as the major factor influencing the salting effect of
carbonyl species (Kroll et al., 2005; Ip et al., 2009), Figure 3 shows modified
Setschenow plots for C5-alkene triols, 2-methyltetrols, and levoglucosan, where log
$(K_{H,w}/K_{H,e})$ values were regressed to the molality of sulfate ion in aerosol liquid water
($c_{\mathrm{sulfate}}$, mol kg$^{-1}$ ALWC). However, log $(K_{H,w}/K_{H,e})$ data deviated from their expected
behavior in the modified Setschenow plot at $c_{\mathrm{sulfate}} > 12$ mol kg$^{-1}$ ALWC, which was
also observed for glyoxal (Kampf et al., 2013). This might be because the ambient
particles did not undergo liquid-liquid phase separation at $c_{\mathrm{sulfate}} > 12$ mol kg$^{-1}$ ALWC,
when the average RH (51.5 ± 15.4%) was lower than the lowest deliquescence RH





(61.8%) of major inorganic salts (e.g., $NH_4NO_3$, $(NH_4)_2SO_4$) in ambient aerosols
(Seinfeld and Pandis, 2016), and the corresponding average concentration of aerosol
liquid water was only $5.31 \pm 4.05$ µg m$^{-3}$. In Figure 3, negative correlations ($p < 0.01$)
are observed at $c_{sulfate} < 12$ mol kg$^{-1}$ ALWC, and the $K_s$ values range from -0.17 to -0.15
kg mol$^{-1}$. Figure S6 shows the regressions between log ($K_{H,w}/K_{H,e}$) of individual polyols
and $c_{sulfate}$ without considering the deviations at high $c_{sulfate}$, and nearly all species
exhibit significant negative correlations ($p < 0.01$). These results indicated the "salting-
in" effects for polyol tracers in northern Nanjing, and to our knowledge the present
study is the first to calculate their $K_{H,e}$ and $K_s$. Although several studies have estimated
Henry's law constants for a variety of polar organic compounds in pure water (e.g.,
polyols and polyacids; Compernolle and Müller, 2014a, b), salting effects should be
considered in describing their gas-particle partitioning in the ambient atmosphere.

The average $K_{H,e}$ values of polyol tracers ($10^{13}$–$10^{15}$ mol m$^{-3}$ atm$^{-1}$) in this study

were several orders of magnitude larger than those of carbonyls derived from ambient
measurements ($10^{10}$–$10^{12}$ mol m$^{-3}$ atm$^{-1}$; Shen et al., 2018) and chamber simulations
(~$10^{11}$ mol m$^{-3}$ atm$^{-1}$; Kroll et al., 2005; Volkamer et al., 2006; Galloway et al., 2009). This
is because low molecular weight carbonyls (e.g., glyoxal) are much more volatile ($p^{o,*}_L >$
$10^{-2}$ atm) than our target polyols (Table S3). According to existing studies, the
minimum concentrations of gas-phase glyoxal and methylglyoxal in Chinese cities
(~0.1 µg m$^{-3}$; Liu et al., 2020) are magnitudes higher than the averages of polyol tracers
in this work, while their particle-phase concentrations are of the same magnitude. The
$K_s$ values of polyol tracers from Figures 3 and S5 (-0.17 – -0.037 kg mol$^{-1}$) are in a
similar range as that of glyoxal (-0.24 – -0.04 kg mol$^{-1}$; Kampf et al., 2013; Shen et al.,
2018; Waxman et al., 2015), indicating that the uptake of different water-soluble



organic compounds might be enhanced by sulfate in a similar manner. However, the
mechanisms of "salting-in" effects are not fully understood. Kampf et al. (2013)
inferred that the enhanced uptake of glyoxal was accompanied by chemical reactions in
the aqueous phase (e.g., hydration and oligomerization), and the interactions between
$SO_4^{2-}$ and glyoxal monohydrate had negative Gibbs free energy of water displacement
(Waxman et al., 2015). The net "salting-in" effect of 1-nitro-2-naphthol in NaF solution
was interpreted by postulating hydrogen bonding (Almeida et al., 1983). A direct
binding of cations to ether oxygens was proposed to be responsible for the increased
solubility of water-soluble polymers (Sadeghi and Jahani, 2012). Due to the complexity
of PM composition, the large gap between $K_{H,e}$ and $K_{H,w}$ cannot be closed by the
"salting-in" effect alone, which is supported by the negative intercepts of linear
regressions in Figure 3. As shown in Figures S7, log $(K_{H,w}/K_{H,e})$ values of polyol tracers
also negatively correlate with the aqueous-phase concentrations of WSOC ($c_{WSOC}$),
given that the plots are more scattered at high $c_{WSOC}$. This dependence might be partly
explained using the "like-dissolves-like" rule, and indicate the importance of
heterogeneous chemistry in the particle phase (Hennigan et al., 2009). No significant
correlation was observed between log $(K_{H,w}/K_{H,e})$ and $NH_4^+$ or $NO_3^-$ concentrations.
Therefore, the bulk WSOC and sulfate ion should play important roles during the
condensation of gas-phase polyols, and further research is warranted to explicitly
explain these effects.
**4 Conclusions and implications**

In this work, concentrations of gas- and particle-phase polyol tracers were

measured simultaneously in northern Nanjing. The temporal variations of individual
compounds were dominated by their particle-phase concentrations. Because receptor-





based models identify and quantify aerosol sources based on inter-sample variability,
gas-particle partitioning of polyol tracers should have little influence on source
apportionment barely using particle-phase data in northern Nanjing. When it comes to
other places (e.g., western US) where the concentration of aerosol liquid water is
extremely low, the influence of gas-particle partitioning will still be a concern.
Similar to southeastern US, the ambient atmosphere in northern Nanjing also
favored the condensation of polyol tracers, which was ascribed to the significant ALWC
in these locations. The large gaps of measured versus predicted $K_{p,OM}$ and $K_H$ implied
that the gas-particle partitioning of polyol tracers could not be depicted using
equilibrium absorptive partitioning model or Henry's law alone.  In addition to the
"salting-in" effect primarily due to the sulfate ions, other aerosol components like bulk
WSOC might also be responsible for increasing the partitioning of polyol tracers into
the condensed phase. So, the results of this study have important implications on the
prediction of gas-particle partitioning of water-soluble organics, and further studies are
required to explain their enhanced aqueous-phase uptake mechanistically. Due to the
hygroscopic properties of highly oxidized organic aerosols, the proposed scheme for
gas-particle partitioning of polyol tracers also partly reveals the discrepancy between
modeled and observed SOA in previous studies. However, several pre-assumptions
(e.g., liquid-liquid phase separation) were made for the proposed gas-particle
partitioning scheme in this work, more research is needed to understand the mixing
state of inorganic salts, organic components, and aerosol liquid water in atmospheric
particles.

***Data availability***





Data used in the writing of this paper is available at the Harvard Dataverse
(https://doi.org/10.7910/DVN/U3IGQR, Qin et al., 2021)

*Author contributions*
MX designed the research. CQ and YG performed the sampling and chemical analysis.
CQ, YM, and MX analyzed the data. CQ and MX wrote the paper with significant
contributions from YW, HL, and QW.

*Competing interests*
The authors declare that they have no conflict of interest.

*Acknowledgements*
This research was supported by the National Natural Science Foundation of China
(NSFC, 41701551). Y. W. was supported by the National Science Foundation
Atmospheric Chemistry Program.

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



Table 1. Statistics for measured and predicted log $K_{p,OM}$ of individual polyol tracers.

| Species | No. of obs. | Log $K^m_{p,OM}$ | | | Log $K^t_{p,OM}$ | | |
|---|---|---|---|---|---|---|---|
| | | Median | Average | Range | Median | Average | Range |
| ***Isoprene SOA tracers*** | | | | | | | |
| C5-alkene triol 1 | 53 | 0.47 | 0.33 ± 0.71 | -1.30 – 2.61 | -3.23 | -3.09 ± 0.44 | -3.70 – -1.63 |
| C5-alkene triol 2 | 63 | 0.19 | 0.15 ± 0.55 | -1.02 – 2.26 | -3.70 | -3.62 ± 0.37 | -4.24 – -2.67 |
| C5-alkene triol 3 | 83 | 0.38 | 0.35 ± 0.68 | -1.86 – 2.25 | -3.01 | -2.90 ± 0.48 | -3.70 – -1.63 |
| 2-Methylthreitol | 101 | -0.13 | -0.12 ± 0.48 | -1.15 – 0.92 | -1.90 | -1.87 ± 0.52 | -2.81 – -0.62 |
| 2-Methylerythritol | 95 | -0.089 | -0.011 ± 0.58 | -1.09 – 2.06 | -1.91 | -1.90 ± 0.51 | -2.81 – -0.62 |
| ***Biomass burning tracer*** | | | | | | | |
| Levoglucosan | 65 | 2.34 | 2.23 ± 0.72 | -0.11 – 3.36 | -0.12 | -0.038 ± 0.59 | -1.00 – 1.29 |
| ***Sugars ang sugar alcohols*** | | | | | | | |
| Meso-erythritol | 31 | 0.84 | 0.87 ± 0.53 | -0.47 – 1.81 | -0.80 | -0.65 ± 0.48 | -1.35 – 0.69 |
| Fructose | 85 | 0.55 | 0.65 ± 0.73 | -0.96 – 3.01 | 1.14 | 1.17 ± 0.62 | 0.015 – 2.72 |
| Mannose | 74 | 0.57 | 0.62 ± 0.71 | -0.94 – 2.81 | 1.23 | 1.28 ± 0.66 | 0.18 – 2.81 |
| Glucose | 88 | 0.41 | 0.42 ± 0.67 | -1.08 – 1.92 | 0.31 | 0.34 ± 0.65 | -0.75 – 1.92 |
| Xylitol | 22 | 0.35 | 0.24 ± 0.54 | -1.23 – 0.97 | 3.43 | 3.37 ± 0.57 | 2.11 – 4.39 |
| Arabitol | 30 | 1.40 | 1.46 ± 0.89 | -0.19 – 4.20 | 3.15 | 3.25 ± 0.77 | 2.05 – 4.81 |
| Manitol | 65 | 1.06 | 1.08 ± 0.63 | -0.35 – 2.53 | 2.31 | 2.33 ± 0.70 | 1.15 – 3.98 |



Table 2. Statistics for log $K_{H,e}$ and log $K_{H,w}$ of individual polyol tracers.

| species | No. of obs. | Log $K_{H,e}$ | | | Log $K_{H,w}$ | | |
|---|---|---|---|---|---|---|---|
| | | Median | Average | Range | Median | Average | Range |
| ***Isoprene SOA tracers*** | | | | | | | |
| C5-alkene triol 1 | 53 | 14.0 | 13.9 ± 0.86 | 11.5 – 16.4 | 7.06 | 7.22 ± 0.50 | 6.53 – 8.87 |
| C5-alkene triol 2 | 63 | 13.7 | 13.6 ± 0.73 | 11.2 – 16.1 | 7.24 | 7.34 ± 0.45 | 6.60 – 8.49 |
| C5-alkene triol 3 | 83 | 13.9 | 13.8 ± 0.85 | 10.6 – 16.1 | 7.31 | 7.43 ± 0.55 | 6.53 – 8.87 |
| 2-Methylthreitol | 101 | 13.4 | 13.3 ± 0.70 | 10.9 – 14.8 | 9.96 | 10.0 ± 0.80 | 8.55 – 11.9 |
| 2-Methylerythritol | 95 | 13.5 | 13.5 ± 0.71 | 11.6 – 15.6 | 9.93 | 9.95 ± 0.79 | 8.55 – 11.9 |
| ***Biomass burning tracer*** | | | | | | | |
| Levoglucosan | 65 | 15.7 | 15.7 ± 0.90 | 13.2 – 17.3 | 13.3 | 13.4 ± 0.56 | 12.4 – 14.6 |
| ***Sugars ang sugar alcohols*** | | | | | | | |
| Meso-erythritol | 31 | 14.5 | 14.4 ± 0.60 | 12.8 – 15.6 | 9.44 | 9.65 ± 0.66 | 8.68 – 11.5 |
| Fructose | 85 | 14.2 | 14.1 ± 0.89 | 11.9 – 16.5 | 14.6 | 14.7 ± 0.84 | 13.1 – 16.8 |
| Mannose | 74 | 14.0 | 14.1 ± 0.94 | 12.1 – 16.8 | 10.9 | 10.9 ± 0.88 | 9.46 – 13.0 |
| Glucose | 88 | 13.9 | 13.9 ± 0.93 | 11.3 – 16.3 | 14.6 | 14.7 ± 0.85 | 13.2 – 16.8 |
| Xylitol | 22 | 13.8 | 13.7 ± 0.72 | 12.6 – 15.0 | 12.1 | 12.1 ± 0.62 | 10.7 – 13.2 |
| Arabitol | 30 | 15.1 | 15.0 ± 1.23 | 13.0 – 18.2 | 11.2 | 11.4 ± 0.83 | 10.0 – 13.0 |
| Mannitol | 65 | 14.6 | 14.5 ± 0.94 | 12.1 – 16.4 | 12.9 | 12.9 ± 1.15 | 11.0 – 15.6 |





# Figure 1

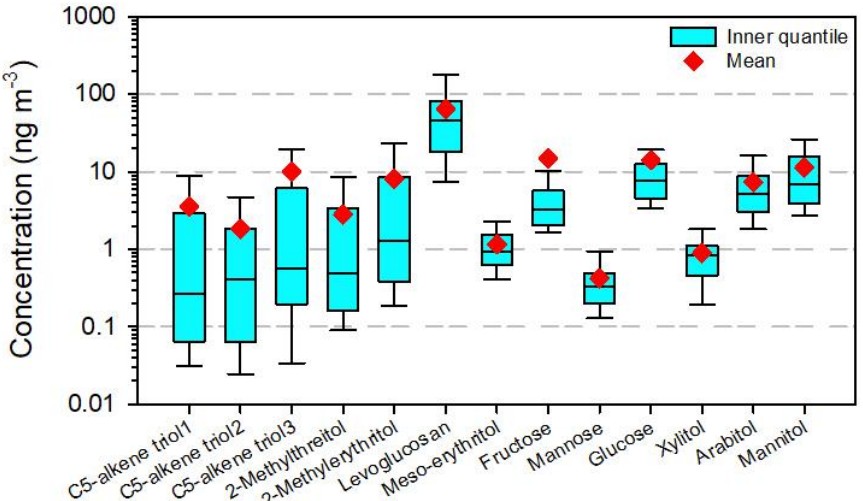

Figure 1. Total concentrations of individual polyols ($Q_f$ + $Q_b$ + PUF) in the ambient atmosphere of northern Nanjing. The boxes depict the median (dark line), inner quantile range (box), 10th and 90th percentiles (whiskers), and the mean (red diamond).





Figure 2

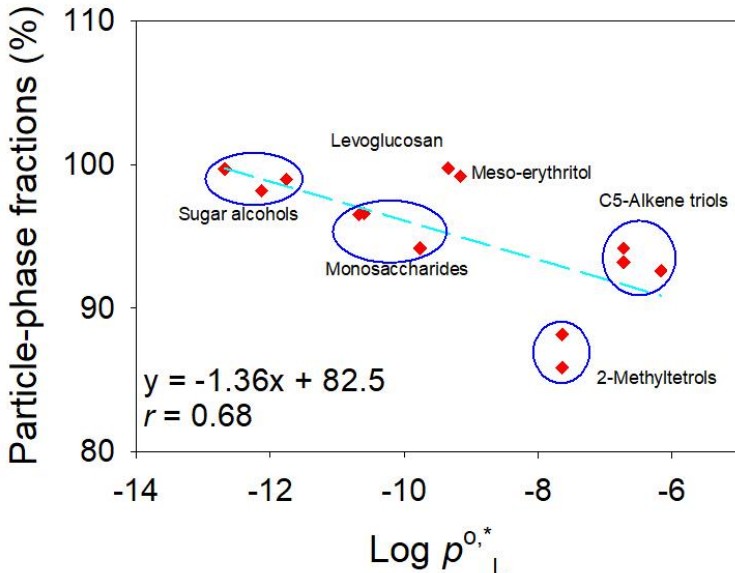

Figure 2. Linear relationship between average aerosol-phase fractions and log $p^{o,*}_{L}$ of polyol tracers.





# Figure 3

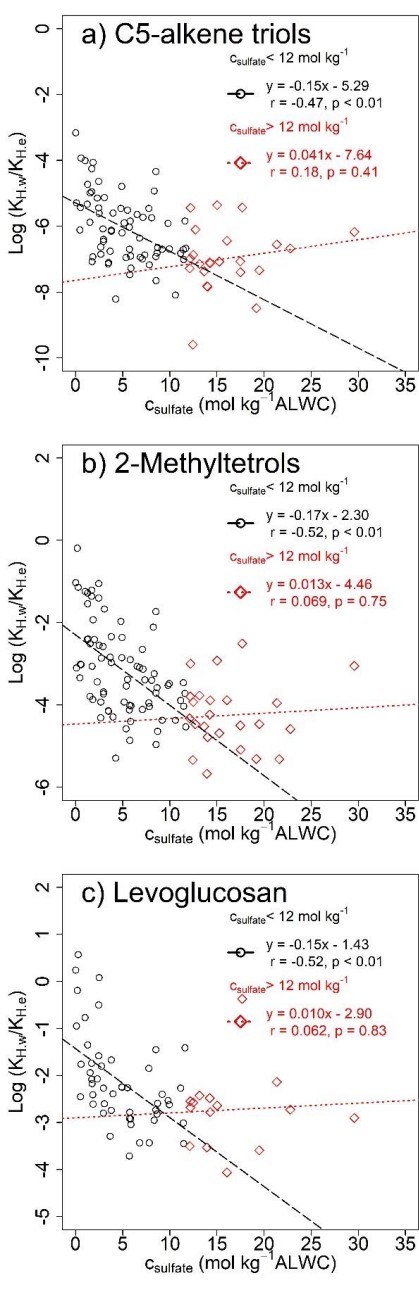

Figure 3. Modified Setschenow plots of log ($K_{H,w}/K_{H,e}$) vs. $c_{sulfate}$ for (a) C5-alkene triols, (b) 2-methyltetrols, and (c) levoglucosan.