# Peer review of "Gas-particle partitioning of polyol tracers in the western Yangtze"

_Atmospheric Chemistry and Physics, 2021_

## Author Comment (AC1)

In this work, the author's present an in-depth look at the gas-particle partitioning of a handful of important, highly oxygenated compounds. Uncertainty in the partitioning of these compounds, and the potential of salting-in to describe observed discrepancies is an important unanswered question that is of interest to readers of this journal. Overall, this work is technically sound and should be published. Some minor concerns and comments are described below:

*Response:*
    Thanks for the reviewer's comments, and we'll reply these point by point.

**General comments:**

**1.** The approach to modeling partitioning nicely accounts for partitioning between phases, but a lot of the information to understand their approach is split between the SI and the main text. Some of the SI I think should maybe be brought into the main text (probably at least Eq. S1 and/or S2, and Figure S2)

*Response:*

    Thanks. We have moved those equations and Figure S2 (now Figure 1) into the main text. To clarify the approach used for the exploration of gas-particle partitioning of polyol tracers, three partitioning cases were defined in the revised manuscript.

Pages 10-13, lines 238-295
"**Calculations of partitioning coefficients**. Here, we defined three partitioning cases to explore the influence of dissolution in aerosol liquid water on gas-particle partitioning of polyol tracers in the atmosphere. *Case 1* presumes instantaneous equilibrium between the gas phase and particulate OM based on the equilibrium absorptive partitioning theory. In this case, particulate OM is assumed to be the only absorbing phase and behave as an ideal solution. Then the absorptive gas-particle partitioning coefficients ($K_{p,OM}$, $m^3\,\mu g^{-1}$) were calculated from measurements ($K^m_{p,OM}$) and predicted theoretically ($K^t_{p,OM}$) as follows

$$K^m_{p,OM} = \frac{F/M_{OM}}{A} \tag{2}$$

$$K^t_{p,OM} = \frac{RT}{10^6 \overline{MW}_{OM}\zeta_{OM}p^o_L} \tag{3}$$

[revised manuscript text omitted]

**2.** A critical question in this work, I think, what is the uncertainty on F%? Uncertainty on these measurements is not really discussed. This parameter is calculated as the ratio of two measurements, each of which likely have at least 10-15% uncertainty (typical of GC), so there is some uncertainty on F% for any given point (though that may decrease as you get to the extreme cases of being mostly in the particle phase as in this case). That doesn't account for the uncertainty on the breakthrough which is significant (e.g., methyltetrols breakthrough is ~20+/-10%, so the correction factor for breakthrough is between a factor of roughly 1.1 and 1.3). Random error on each point should be reduced in the average (i.e., the average F% is known better than any one point), but the averages could be susceptible to systemic errors like uncertainty in

breakthrough that could create bias. I do notice that during the periods of high particle concentrations for e.g., 2-MTs, F% does sometimes reach 100%, so perhaps bias is minimal, but it would be nice to get some estimates of uncertainties, or additional discussion and analysis of potential error. For instance, couldn't the divergence in Figure 2 of the isoprene tracers from the line be due to some bias like uncertainty in breakthrough?

*Response:*

Thanks. The measurement uncertainties will not only impact the calculation results of particle phase fractions ($F\%$) of polyol tracers, but also the partitioning coefficients of gas vs. particulate organic ($K^m_{p,OM}$) and aqueous ($K^m_{H,e}$) phases.

The uncertainty estimation for measurements and partitioning coefficients were added in the revised manuscript and supplementary information, respectively.

Page 13, lines 296 – 303 of the main text:
"***Uncertainty estimation.*** To obtain the uncertainty associated with the calculation of $F\%$ and partitioning coefficients ($K^m_{p,OM}$ and $K^m_{H,e}$), measurement uncertainties of polyol tracers in filter and PUF samples were estimated from their recoveries and breakthrough for gaseous sampling. The root sum of squares (RSS) method was applied to propagate uncertainties of gas and particle-phase concentrations for $F\%$, $K^m_{p,OM}$, and $K^m_{H,e}$ calculations. Details of the uncertainty estimation and propagation methods were provided in Text S5, and the average relative uncertainties were summarized in Table S5."

Text S5 in supplementary information:
"***Text S5. Uncertainty estimation methods***
In this work, the measurement results of some polyol tracers in filter and PUF samples are subject to substantial uncertainties due to their low and variable recoveries (Table S2) and excessive breakthrough (Figure S2). A general equation was derived to estimate measurement uncertainties of individual polyols in filter and PUF samples

$$\Delta C = \sqrt{(\text{error fraction} \times \text{concentration})^2 + (0.5 \times \text{detection limit})^2} \qquad (5)$$

where $\Delta C$ is the uncertainty of target species in filter ($\Delta Q_f$ and $\Delta Q_b$, ng m$^{-3}$) or PUF ($\Delta$PUF, ng m$^{-3}$) samples. The error fraction (%) of filter sample analysis was defined as half of the difference between maximum and minimum recoveries scaled by the average (Table S2), which was divided by (1 - average breakthrough) for PUF analysis (Figure S2). The average breakthrough of meso-erythritol (23.8%), mannose (38.1%), xylitol (36.4%), and arabitol (36.4%) were set as those of C5-alkene triols, glucose, and mannitol, respectively. According to the gas-particle separation method in this work, $\Delta Q_f$ was used to represent the uncertainty of particle-phase concentration ($\Delta F$, ng m$^{-3}$), and the uncertainty of gas-phase concentration ($\Delta A$, ng m$^{-3}$) was propagated by

$$\Delta A = \sqrt{\Delta Q_b^2 + \Delta\text{PUF}^2} \qquad (6)$$

Then the uncertainty of total concentration ($\Delta S$, ng m$^{-3}$) was calculated as
$$\Delta S = \sqrt{\Delta F^2 + \Delta A^2} \qquad (7)$$
The uncertainties of particle-phase fractions ($\Delta F\%$) and partitioning coefficients ($K^m_{p,OM}$ and $K^m_{p,WIOM}$, m$^3$ ug$^{-1}$; $K^m_{H,e}$, mol m$^{-3}$ atm$^{-1}$) were estimated by propagating

$\Delta F$, $\Delta S$, and $\Delta A$ using a simplified root sum of squares (RSS) method (Dutton et al., 2009)

$$\Delta F\% = \sqrt{\left(\frac{\partial F\%}{\partial F}\Delta F\right)^2 + \left(\frac{\partial F\%}{\partial S}\Delta S\right)^2} \times 100\% \qquad (8)$$

$$\Delta K = \sqrt{\left(\frac{\partial K}{\partial F'}\Delta F'\right)^2 + \left(\frac{\partial K}{\partial A}\Delta A\right)^2} \qquad (9)$$

where $\Delta K$ is the uncertainty of $K^{m}_{p,OM}$, $K^{m}_{p,WIOM}$, or $K^{m}_{H,e}$; $F'$ could be $F$, concentrations of polyols in WIOM ($F_{WIOM}$) or aqueous ($F_{w}$) phases, depending on the partitioning scheme (*Cases 1–3*) and partitioning coefficient for calculation. $\Delta F$ was split into $\Delta F_w$ and $\Delta F_{WIOM}$ (or $\Delta F_{OM}$) based on their ratios in eq. 4 of the main text. In Table S5, the estimated uncertainties are summarized and expressed in average ratios. As $K^{m}_{p,OM}$ and $K^{m}_{H,e}$ are all directly related to the ratio of particle- ($F$, ng m$^{-3}$) and gas-phase ($A$, ng m$^{-3}$) concentrations (eqs. 2, 4, 5, and 6 in the main text), their average $\Delta K/K$ values are the same (Table S5)."

Moreover, Figure 2 (now Figure 3 in the revised manuscript, shown below) has been changed by including the uncertainty of $F\%$ and one standard deviation of log $p^{o,*}_{L}$ derived from different estimation tools (Table S4). Because monosaccharides (e.g., fructose) and sugar alcohols (e.g., mannitol) had low and variable recoveries (Table S2) and excessive breakthrough for gaseous sampling (Figure S2), their average $F\%$ uncertainties (6.16–31.2%) are much larger than those of isoprene SOA tracers and levoglucosan (3.33–7.24%). As shown in the figure below, more than 95% of polyols with extremely low vapor pressures ($< \sim10^{-10}$ atm) are distributed into the particle phase, so their prominent uncertainties in $F\%$ are not physically meaningful. Considering the uncertainties in $F\%$ and log $p^{o,*}_{L}$ and high average $F\%$ ($> 85\%$) of polyol tracers, a dependence of $F\%$ on the vapor pressure could not be determined.

[Figure]

**Figure 3.** Average particle-phase fractions and log $p^{o,*}_{L}$ of individual polyol tracers. Whiskers represent uncertainties of $F\%$ and one standard deviation of log $p^{o,*}_{L}$ derived from different estimation tools.

In the revised manuscript, we added the above discussions on uncertainties of $F\%$ (Pages 17-18, lines 410–419).

"In Figure 3, The average $F\%$ uncertainties (6.16–31.2%) of monosaccharides (e.g., fructose) and sugar alcohols (e.g., mannitol) were larger than those of isoprene SOA tracers and levoglucosan (3.33–7.24%) due to their low and variable recoveries (Table S2) and excessive breakthrough (Figure S2). However, the estimated uncertainties of $F\%$ for less volatile polyols ($p^{o,*}_L <$ ~$10^{-10}$ atm) were not physically meaningful, as more than 95% of these compounds existed in the particle phase. Considering the uncertainties in $F\%$ and log $p^{o,*}_L$ and high average $F\%$ ($> 85\%$) of target polyol tracers, a dependence of $F\%$ on the vapor pressure could not be determined, and the seasonality and day-night difference ($p > 0.05$) of $F\%$ were obscured."

**3.** Some of these results might be impacted by uncertainty in theoretical partitioning coefficients and by poorly constrained empirically determined coefficients.
All of the tracers shown here have values close to 100%, so there is not much dynamic range in the data and it might be susceptible to biases.
Calculation of partitioning coefficients in particular could be sensitive to uncertainties because of this (as you approach 100% in one phase, small changes in partitioning might imply large changes in K). For this reason, I'm not sure the discussion of comparison between theoretical and measured K is always that meaningful. As an extreme example, levoglucosan looks like it is always at ~100% except for maybe one point. Under these conditions, how can any meaningful K be measured, since a partitioning coefficient of 10^3 and 10^100 would both produce the same effect?
On the other hand, there is substantial error in the theoretical values as well, with uncertainty in vapor pressure likely on the order of 1-2 orders of magnitude for most of these compounds (and some evidence that EPI has a tendency to overestimate compared, see Barley and McFiggans 2010).
While I agree that the time-dependent comparisons between measured and expected K against things like sulfate (e.g., Figure 3) provide insight, comments like that on line 345 comparing measured to theoretical K quantitatively aren't that meaningful.
Similarly, if you account for these sources of uncertainty, it's not clear to me that the lines in Figure 3 necessarily have a negative intercept as described. Some discussion of these uncertainties and biases might help clarify what we do know (e.g., these tracers are mostly particle phase, theoretical vapor pressures are wrong, and correlations with absorptive theory are poor), with the quantitative aspects we don't know as well (e.g., how wrong are the vapor pressures, how strong is the dependence on salt).

*Response:*

In our replies to the previous two comments, we have clarified the three partitioning cases defined in this work and uncertainty estimation methods for measurements and measurement-based partitioning coefficients. To address the uncertainty of vapor pressure estimated using EPI, a variety of estimation tools were deployed to provide a reasonable vapor pressure range for each polyol tracer. Referring to Table S4 of the revised supplementary information, we can find that the vapor pressure ranges of isoprene SOA tracers, levoglucosan, and meso-erythritol are within two orders of magnitude, while those for monosaccharides and mannitol are larger ($> 10^3$).

The target for comparing measured and predicted partitioning coefficients is to demonstrate that particulate OM is not the only absorbing phase, and the aerosol liquid water also plays a significant role in influencing the gas-particle partitioning of polyol tracers. In Table S5 of the revised supplementary information, the average relative uncertainties of measurement-based partitioning coefficients range from a few percent to ~50%, which will result in an uncertainty of less than ± 0.3 for logarithms of partitioning coefficients. In revised Table 1, the measurement-based $K_{p,OM}$ ($K^m_{p,OM}$) of isoprene SOA tracers, levoglucosan, and meso-erythritol for *Case 1* were more than 10 times greater than most theoretical predictions ($K^t_{p,OM}$). When the dissolution in aerosol liquid water was considered, their average log $K^m_{p,OM}$ became much closer to or even lay within the range (e.g., levoglucosan) of log $K^t_{p,OM}$, whenever water-soluble and water-insoluble organic matter (OM) partitioned into separate (*Cases 2*) or single (*Case 3*) liquid phases. These results support that the partitioning between gas and aerosol liquid water should not be ignored for water-soluble organic compounds like polyol tracers.

In other words, if particulate OM is the only absorbing phase in aerosols, log $K^m_{p,OM}$ values of *Case 1* will be closer to log $K^t_{p,OM}$ ranges, and the expected *F%* of isoprene SOA tracers, levoglucosan, and meso-erythritol should be much lower than measured in this work. To make the comparison between measured and predicted $K_{p,OM}$ meaningful, we predefined three partitioning cases and estimate vapor pressures of polyol tracers using several tools, and compared average log $K^m_{p,OM}$ of different partitioning cases with log $K^t_{p,OM}$ ranges. Moreover, the $K^m_{p,OM}$ uncertainty derived from measurements was estimated and would not impact the main conclusion.

The negative intercepts shown in Figures 4 and S5 of the revised manuscript and supplements are not likely ascribed to the small relative uncertainties of partitioning coefficients (Table S5). Shen et al. (2018) also identified negative intercepts of linear regressions between log ($K_{H,w}/K_{H,e}$) and $c_{sulfate}$ for glyoxal and methylglyoxal in the ambient atmosphere, and attributed this to unknown gas-particle partitioning mechanisms. By performing chamber experiments and comparing to existing laboratory studies, Kampf et al. (2013) found that the exponential increase of gas vs. aqueous phase partitioning coefficient ($K_{H,e}$, mol m$^{-3}$ atm$^{-1}$) with sulfate concentration only occurred at $c_{sulfate}$ < 12 mol kg$^{-1}$ ALWC. In Figures 4 and S5, the log ($K_{H,w}/K_{H,e}$) values of most polyols increase faster as $c_{sulfate}$ approaches 0, supporting that the enhanced uptake at low sulfate concentrations could be partly parameterized by the equation defining "salting-in" effects. However, the "salting-in" effect is a known phenomenon that is not likely linked with a specific physical or chemical mechanism. In previous studies, the increased partitioning of polar organic compounds to the particle phase was often attributed to organic-inorganic interactions, including reactive uptake, aqueous phase chemistry, etc.

In addition to the changes mentioned in previous two responses, we reorganized and rewrote the discussions on partitioning coefficients of gas versus organic and aqueous phases (*Sections 3.4 and 3.5*), considering the uncertainties in both measurements and predictions.

Pages 18-24, lines 420-560

[revised manuscript text omitted]

*Specific comments:*

**4.** Line 57: Typo, "Filed" should say "Field"

*Response:*

It has been revised as suggested. (Page 4, line 80)

**5.** Line 85: Probably also worth mentioning that Yatavelli et al. (2014, 10.5194/acp-14-1527-2014) and Isaacman-VanWertz et al. (2016) also found good agreement with theory for alkanoic acids

*Response:*

The two references were added, and the original expression has been changed into
"Unlike non-polar species (e.g., *n*-alkanes, polycyclic aromatic hydrocarbons) and alkanoic acids that are well simulated (Simcik et al., 1998; Xie et al., 2014a; Yatavelli et al., 2014; Isaacman-VanWertz et al., 2016), particle-phase concentrations of carbonyls were underestimated by several orders of magnitude when particulate OM is presumed as the only absorbing phase in ideal condition (Healy et al., 2008; Kampf et al., 2013; Shen et al., 2018)." (Page 5, 109-114)

**6.** Line 90: Typo, "every" should say "very"

*Response:*

    It has been revised as suggested. (Page 5, line 117)

**7.** Line 176: When quantifying the isoprene tracers using meso-erythritol, I assume the effect of fragmentation on the quant ion was accounted for, but that is not clear here or in the supplmental. In other words, while m/z 219 is used for the 2-MTs, something different, likely m/z 217, is used for meso-erythritol, and the fraction of total signal that is the quant ion could be different - if this was corrected for it should be stated at least in the SI, if not it should be justified.

*Response:*

    In the current work, we did not consider the effect of fragmentation on the quant ion due to the following reasons:
(1) Meso-erythritol and isoprene tracers (C5-alkene triols and 2-methyltetrols) should have different total ion signal intensity for the same amount, so the calibration curve of meso-erythritol is expected to differ from isoprene tracers. Even if the effect of fragmentation on the quant ion was accounted for, the quantification accuracy might not be improved. A number of previous studies used meso-erythritol as surrogate for isoprene tracers without considering the effect of fragmentation (Claeys et al., 2004; Hu et al., 2008; Ding et al., 2008, 2012; Lin et al., 2013; Xie et al., 2014). It should also be noted that ketopenic acid was used to quantify all SOA tracers by Kleindienst et al. (2007).
(2) The present study focused on the gas-particle partitioning of polyol tracers. Their particle-phase fractions, measurement-based partitioning coefficients of gas versus organic ($K^m_{p,OM}$) and aqueous ($K^m_{H,e}$) phases were all directly related to the ratio of particle- to gas-phase concentrations, which is not impacted by the effect of fragmentation.
(3) The total ion peaks corresponding to isoprene tracers in ambient air samples usually combined with other compounds, thus the quant ions (m/z 219) fractions of 2-methyltetrols in total ion signal (0.14 ± 0.035, 0.20 ± 0.015) had larger variability than that of meso-erythritol (0.12 ± 0.0039), then the correction for fragmentation effect might introduce new uncertainty.

    In the revised supplementary information, we added some justifications for not correcting the fragmentation effect.
    "Meso-erythritol and isoprene SOA tracers were expected to have different total ion intensity for the same amount, and the total ion signals of isoprene tracers in ambient air sample often co-eluted with other compounds, so the fragmentation difference between quantification ions of meso-erythritol (m/z 217) and isoprene SOA tracers (m/z 219) was not adjusted in this work. It should be noted that meso-erythritol was used as the surrogate for isoprene tracers without considering the fragmentation effect in a number of previous studies (Claeys et al., 2004; Hu et al., 2008; Ding et al., 2008, 2012; Lin et al., 2013; Xie et al., 2014)."

**8.** Line 197: If gas-phase mass is being taken as Qb plus PUF, what is the purpose of the Qb measurement at all? Why not just do Qf backed by a PUF?

*Response:*

The backup quartz fiber filter was typically used to assess positive sampling artifact of particulate OC due to gaseous adsorption (Chow et al., 2010; Subramanian et al., 2004). In a companion study to this work, concentrations of bulk species in $PM_{2.5}$ were determined by subtracting measurement results of $Q_b$ from $Q_f$ (Yang et al., 2021). This has been mentioned in section 2.2 of the manuscript (Pages 8, lines 176-178).

Concentrations of polyol tracers detected on $Q_b$ reflected filter-based sampling artifacts, which is ascribed to gaseous adsorption ("positive artifact") and evaporation loss from particles on $Q_f$ ("negative artifact"). If polyol tracers on $Q_b$ account for significant fractions of their total concentrations, then the separation of their particle- and gas-phase concentrations will depend largely on sampling artifact corrections. In the current work, polyol tracers were predominantly observed on $Q_f$ with averages 1-3 orders of magnitude higher than those on $Q_b$ and PUF, and it would be safe to assume that $Q_b$ values were equally contributed by positive and negative artifacts. So, $Q_b$ measurement is necessary in determining the magnitude of sampling artifacts.

The original expressions have been revised (Pages 9-10, lines 218-229).

"Polyol tracers detected in $Q_b$ samples are contributed by both gaseous adsorption ("positive artifact") and particle-phase evaporation from $Q_f$ samples ("negative artifact"), but their relative contributions are unknown. Xie et al. (2014b) adjusted particle- and gas-phase concentrations of levoglucosan and 2-methyltetrol based on $Q_b$ measurements in two different ways. One assumed that $Q_b$ values were completely attributed to gaseous adsorption; the other presumed equal contributions from gaseous adsorption and $Q_f$ evaporation. However, negligible difference in gas-particle distribution was observed due to the small $Q_b$ values. In Table S3, concentrations of polyol tracers on $Q_b$ are far below those on $Q_f$, and it would be safe to presume equal positive and negative artifacts. In this study, particle-phase concentrations of polyols were represented by $Q_f$ values, and the gas phase was calculated as the sum of $Q_b$ and PUF measurements."

**9.** Line 220-225: I'm not completely clear on the partitioning approach. The Kow is used to partition the particle fraction between condensed phases - is this information then used in the gase-particle partitioning? For instance, is K_OM used for the organic component, and K_H used for the aqueous? I guess I'm just not clear on how Kow fits into the scheme.

*Response:*

Yes, the $K_{OW}$ is used to calculate concentrations of polyol tracers in two separate aerosol phases (organic and aqueous). $K_{p,OM}$ and $K_H$ represent gas-particle partitioning coefficients of gas versus organic and aqueous phases, respectively.

To make it clear, we defined three partitioning cases, and elucidate the calculation methods for the two partitioning coefficients in the revised manuscript.

Pages 10-13, lines 238-295 (See responses to *Comment 1*, eqs 4, 5, and 6)

**10.** Line 297-298: Isaacman-VanWertz et al. (2016) also show hourly diurnal profiles with a daytime high, it is interesting to see in this work that the difference between daytime and nighttime values was not significant. As noted later in the manuscript, in the summer when there are actually strong concentrations of these compounds, there is a strong diurnal, so I'm not sure it really makes sense to claim here there is no diurnal.

*Response:*

Similar to the observations by Fu and Kawamura (2011) at a forest site in Hokkaido, Japan, concentrations of isoprene SOA tracers in southeastern US and central Amazonia also exhibited peak concentrations from late afternoon to mid night. In the current work, daytime and nighttime samples were collected during 8:00 AM – 7:00 PM and 7:00 PM – 7:00 AM (next day), respectively. Neither the daytime or nighttime sample covered the whole period when isoprene SOA had peak concentrations. This explains why the strong diurnal variations of isoprene SOA tracers were not captured in this work.

The original expression was changed into
"Previous field studies observed strong diurnal variations of isoprene SOA tracers with peak concentrations from afternoon till midnight (Fu and Kawamura, 2011; Isaacman-VanWertz et al., 2016). Although no IEPOX will be generated from the oxidation of isoprene by •OH and $HO_2$• after sunset, the formations of C5-alkene triols and 2-methyltetrols might continue until pre-existing IEPOX is exhausted. In this work, neither the daytime (8:00 AM–7:00 PM) or night-time (7:00 AM–7:00 AM next day) sample covered the whole period when isoprene SOA tracers had peak concentrations, and the strong diurnal variations of C5-alkene triols and 2-methyltetrols were not captured." (Page 16, 367-375)

**11.** Line 336: This is the first place F% is given any error or range, though I assume here the uncertainty is the standard deviation.

*Response:*

We clarified in the revised manuscript that the numbers here represented ranges of average ± standard deviation. (Page 17, lines 405)
Moreover, measurement uncertainties and their influences on $F$% and partitioning coefficients were estimated in the revised manuscript. (Text S5 and Figure 3, see responses to *Comment 2*; Table S5, shown below)

Table S5. Average relative uncertainties of measurements and calculated parameters.

| Species | $\Delta F/F$[a] | $\Delta A/A$[b] | $\Delta S/S$[c] | $\Delta F\%/F\%$[d] | $\Delta K/K$[e] |
|---|---|---|---|---|---|
| *Isoprene SOA tracers* | | | | | |
| C5-alkene triol 1 | 0.028 | 0.032 | 0.027 | 0.037 | 0.043 |
| C5-alkene triol 2 | 0.028 | 0.054 | 0.033 | 0.036 | 0.059 |
| C5-alkene triol 3 | 0.028 | 0.077 | 0.034 | 0.038 | 0.084 |
| 2-Methylthreitol | 0.028 | 0.051 | 0.028 | 0.033 | 0.059 |
| 2-Methylerythritol | 0.028 | 0.066 | 0.030 | 0.035 | 0.072 |
| *Biomass burning tracer* | | | | | |
| Levoglucosan | 0.051 | 0.16 | 0.054 | 0.072 | 0.17 |
| *Sugars ang sugar alcohols* | | | | | |
| Meso-erythritol | 0.028 | 0.11 | 0.028 | 0.040 | 0.12 |
| Fructose | 0.23 | 0.27 | 0.26 | 0.31 | 0.36 |
| Mannose | 0.045 | 0.27 | 0.049 | 0.062 | 0.28 |
| Glucose | 0.094 | 0.28 | 0.10 | 0.18 | 0.31 |
| Xylitol | 0.10 | 0.12 | 0.10 | 0.14 | 0.16 |
| Arabitol | 0.097 | 0.26 | 0.099 | 0.14 | 0.28 |
| Mannitol | 0.21 | 0.42 | 0.21 | 0.29 | 0.47 |

[a] Particle-phase concentration; [b] gas-phase concentration; [c] total concentration; [d] particle-phase fraction; [e] partitioning coefficients of gas vs. organic and aqueous phases.

**12.** Line 370: This could also be due to some systematic bias in how EPI estimates vapor pressures. This might be reconcile by testing other vapor pressure estimation methods. For example EVAPORATION agrees with EPI on the vapor pressure of erythritol (log vp (atm) = ~ -8.2-8.5) but EPI estimates mannose to be two orders of magnitude more volatile (log vp (atm) = -9.5) than EVAPORATION estimates (log vp (atm) = -11.5). This also highlights the uncertainty of using comparisons against theoretical K to draw conclusions.

*Response:*

To address the concern on uncertainties of vapor pressures, other estimation tools were tested (Table S4). The revised Table 1 (shown below) compared measurement-based log $K_{p,OM}$ of predefined *Cases 1-3* and theoretical predictions based on different vapor pressure estimation methods. Then the discussions on comparisons between measured and predicted $K_{p,OM}$ were revised accordingly (Section 3.4, see responses to *Comment 3*; pages 18-20, lines 420-480).

As shown in Table 1, when the solubility in aerosol liquid water is considered (*Cases 2* and *3*), the agreement between log $K^{m}_{p,OM}$ and log $K^{t}_{p,OM}$ for isoprene SOA tracers and levoglucosan has been improved substantially. While the deviations of log $K^{m}_{p,OM}$ versus log $K^{t}_{p,OM}$ for monosaccharides and sugar alcohols became larger. This is expected to be caused by the overestimations of their gas phase concentrations due to sampling artifacts, since the vapor pressures of sugar polyols were orders of magnitude lower than isoprene SOA tracers and levoglucosan.

Table 1. Comparisons of measurement-based log $K_{p,OM}$ (m$^3$ μg$^{-1}$) at three proposed cases and predicted values.

| Species | No. of obs. | Log $K^m_{p,OM}$ [a] | | | Log $K_{p,OM}$ [b] | | | |
|---|---|---|---|---|---|---|---|---|
| | | Case 1 | Case 2 | Case 3 | EPI | EVAPORATION | SPARC | SIMPOL |
| ***Isoprene SOA tracers*** | | | | | | | | |
| C5-alkene triol 1 | 53 | 0.33 ± 0.71 | -0.79 ± 0.86 | -0.82 ± 0.85 | -3.09 | -2.84 | -1.19 | -2.88 |
| C5-alkene triol 2 | 63 | 0.15 ± 0.55 | -1.02 ± 0.74 | -1.05 ± 0.73 | -3.62 | -3.67 | -4.14 | -2.85 |
| C5-alkene triol 3 | 83 | 0.35 ± 0.68 | -0.83 ± 0.86 | -0.86 ± 0.85 | -2.90 | -2.65 | -1.00 | -2.69 |
| 2-Methylthreitol | 101 | -0.12 ± 0.48 | -2.09 ± 0.71 | -2.09 ± 0.70 | -1.87 | -1.30 | -1.18 | -0.47 |
| 2-Methylerythritol | 95 | -0.011 ± 0.58 | -1.96 ± 0.71 | -1.96 ± 0.71 | -1.90 | -1.34 | -1.22 | -0.50 |
| ***Biomass burning tracer*** | | | | | | | | |
| Levoglucosan | 65 | 2.23 ± 0.72 | 0.63 ± 0.90 | 0.62 ± 0.90 | -0.04 | -0.81 | 1.04 | -0.76 |
| ***Sugars ang sugar alcohols*** | | | | | | | | |
| Meso-erythritol | 31 | 0.87 ± 0.53 | -1.43 ± 0.60 | -1.43 ± 0.60 | -0.65 | -1.21 | -0.45 | |
| Fructose | 85 | 0.65 ± 0.73 | -1.20 ± 0.83 | -1.20 ± 0.89 | 1.17 | 2.76 | 6.94 | |
| Mannose | 74 | 0.62 ± 0.71 | -2.12 ± 0.95 | -2.12 ± 0.95 | 1.28 | 2.13 | 4.77 | |
| Glucose | 88 | 0.42 ± 0.67 | -2.77 ± 0.93 | -2.77 ± 0.93 | 0.34 | 3.75 | 7.32 | |
| Xylitol | 22 | 0.24 ± 0.54 | -2.61 ± 0.72 | -2.61 ± 0.72 | 3.37 | 2.34 | 3.57 | |
| Arabitol | 30 | 1.46 ± 0.89 | -1.35 ± 1.24 | -1.35 ± 1.24 | 3.25 | 1.67 | 2.90 | |
| Manitol | 65 | 1.08 ± 0.63 | -2.24 ± 0.95 | -2.24 ± 0.95 | 2.33 | 4.16 | 6.68 | |

[a] Average ± standard deviation; [b] temperature range: -4~36 °C.

**13.** Line 391: Here and throughout, why use K_H,e and K_H,w for measured and predicted, respectively, for Henry's law, but K_p^m and K_p^t or measured and predicted, respectively, for absorptive partitioning. I think the discussion would be more clear if notation were more consistent (e.g, K_H^m and K_H^t

*Response:*

Thanks. To make it more clear and consistent, $K_{H,e}$ and $K_{H,w}$ were changed into $K^m_{H,e}$ and $K^t_{H,w}$ if necessary throughout the manuscript.

14. Line 407: out of curiosity, why did the authors choose to switch to molality instead of molarity?

*Response:*

The switch from molarity to molality will not influence the correlation between log ($K_{H,w}/K_{H,e}$) and sulfate concentrations. It also makes it easier to compare the salting constant ($K_s$) with previous studies (Kampf et al., 2013; Waxman et al., 2015; Shen et al., 2018).

---

## Author Comment (AC2)

This study presents the results of simultaneous gas- and particle-phase measurements of oxygenated organic compounds in Nanjing, China. Due to some potential measurement artifacts with some compounds, the results focus on C5-alkene triols, 2-methyltetrols, and levoglucosan. The major finding is that the particle-phase fraction of these compounds were on average orders of magnitude higher than can be explained by either absorptive partitioning theory or Henry's law. There is moderate evidence that sulfate caused a "salting in" effect, though more discussion and/or data are needed to support this point (comment detailed below). The writing and organization are generally good, and the topic is of interest to a broad audience. I recommend the manuscript for publication after the following comments are addressed.

*Response:*

Thanks for the reviewer's comments, and we will reply these point by point in the reviewers' specific comments.

**Specific Comments**
1. How much does measurement uncertainty affect the partitioning coefficients? The stated acceptable threshold for breakthrough (< 33%) seems relatively high. How do the observed levels of breakthrough contribute to the uncertainty in the partitioning coefficients? Further, did breakthrough vary as a function of ambient temperature, OA loading, etc? Discussion of these points is needed.

*Response:*

The estimation of measurement uncertainties and their influences on partitioning coefficients were added in the revised manuscript. In general, measurement uncertainties of polyol tracers in filter samples were estimated from averages and ranges of their recoveries and method detection limits. Breakthrough of gaseous sampling was additionally considered for measurement uncertainties of PUF samples. Uncertainties associated with the calculations of particle-phase fractions and partitioning coefficients were estimated using a simplified root sum of squares (RSS) method by propagating measurement uncertainties of gas- and particle-phase concentrations. Details of the method were provided in Text S5 in supplementary information and mentioned in the main text.

Text S5 in supplementary information:
"***Text S5. Uncertainty estimation methods***
In this work, the measurement results of some polyol tracers in filter and PUF samples are subject to substantial uncertainties due to their low and variable recoveries (Table S2) and excessive breakthrough (Figure S2). A general equation was derived to estimate measurement uncertainties of individual polyols in filter and PUF samples

$$\Delta C = \sqrt{(\text{error fraction} \times \text{concentration})^2 + (0.5 \times \text{detection limit})^2} \qquad (5)$$

where $\Delta C$ is the uncertainty of target species in filter ($\Delta Q_f$ and $\Delta Q_b$, ng m$^{-3}$) or PUF ($\Delta$PUF, ng m$^{-3}$) samples. The error fraction (%) of filter sample analysis was defined as half of the difference between maximum and minimum recoveries scaled by the average (Table S2), which was divided by (1 - average breakthrough) for PUF analysis (Figure S2). The average breakthrough of meso-erythritol (23.8%), mannose (38.1%), xylitol (36.4%), and arabitol (36.4%) were set as those of C5-alkene triols, glucose, and

mannitol, respectively. According to the gas-particle separation method in this work, $\Delta Q_f$ was used to represent the uncertainty of particle-phase concentration ($\Delta F$, ng m$^{-3}$), and the uncertainty of gas-phase concentration ($\Delta A$, ng m$^{-3}$) was propagated by

$$\Delta A = \sqrt{\Delta Q_b{}^2 + \Delta PUF^2} \qquad (6)$$

Then the uncertainty of total concentration ($\Delta S$, ng m$^{-3}$) was calculated as

$$\Delta S = \sqrt{\Delta F^2 + \Delta A^2} \qquad (7)$$

The uncertainties of particle-phase fractions ($\Delta F\%$) and partitioning coefficients ($K^m_{p,OM}$ and $K^m_{p,WIOM}$, m$^3$ ug$^{-1}$; $K^m_{H,e}$, mol m$^{-3}$ atm$^{-1}$) were estimated by propagating $\Delta F$, $\Delta S$, and $\Delta A$ using a simplified root sum of squares (RSS) method (Dutton et al., 2009)

$$\Delta F\% = \sqrt{\left(\frac{\partial F\%}{\partial F}\Delta F\right)^2 + \left(\frac{\partial F\%}{\partial S}\Delta S\right)^2} \times 100\% \qquad (8)$$

$$\Delta K = \sqrt{\left(\frac{\partial K}{\partial F'}\Delta F'\right)^2 + \left(\frac{\partial K}{\partial A}\Delta A\right)^2} \qquad (9)$$

where $\Delta K$ is the uncertainty of $K^m_{p,OM}$, $K^m_{p,WIOM}$, or $K^m_{H,e}$; $F'$ could be $F$, concentrations of polyols in WIOM ($F_{WIOM}$) or aqueous ($F_w$) phases, depending on the partitioning scheme (*Cases 1–3*) and partitioning coefficient for calculation. $\Delta F$ was split into $\Delta F_w$ and $\Delta F_{WIOM}$ (or $\Delta F_{OM}$) based on their ratios in eq. 4 of the main text. In Table S5, the estimated uncertainties are summarized and expressed in average ratios. As $K^m_{p,OM}$ and $K^m_{H,e}$ are all directly related to the ratio of particle- ($F$, ng m$^{-3}$) and gas-phase ($A$, ng m$^{-3}$) concentrations (eqs. 2, 4, 5, and 6 in the main text), their average $\Delta K/K$ values are the same (Table S5). "

Page 13, lines 296 – 303 of the main text:
"***Uncertainty estimation.*** To obtain the uncertainty associated with the calculation of $F\%$ and partitioning coefficients ($K^m_{p,OM}$ and $K^m_{H,e}$), measurement uncertainties of polyol tracers in filter and PUF samples were estimated from their recoveries and breakthrough for gaseous sampling. The root sum of squares (RSS) method was applied to propagate uncertainties of gas and particle-phase concentrations for $F\%$, $K^m_{p,OM}$, and $K^m_{H,e}$ calculations. Details of the uncertainty estimation and propagation methods were provided in Text S5, and the average relative uncertainties were summarized in Table S5."

According to the equation for breakthrough calculation, a value of 33% means that the amount of a certain compound in backup PUF samples is half of that in front PUF samples. It was used as an indicator of excessive breakthrough in several previous studies (Peters et al., 2000; Ahrens et al., 2011; Xie et al., 2014a, b). However, the breakthrough value was rarely used to correct measurement results.

In the revised manuscript, the breakthrough of gaseous sampling was included for the estimation of measurement uncertainties and their influences on partitioning coefficients, but its individual contributions cannot be separated. In Table S5 (shown below), uncertainties of gas- and particle-phase concentrations, particle-phase fractions, and partitioning coefficients are expressed in average relative abundance.

Due to the limit in sample number for breakthrough tests and low detection rates, we can hardly evaluate the dependence of breakthrough on ambient temperature or OA loadings. The breakthrough of an ideal sampling method is expected to be extremely low (e.g., <10%) and have no dependence on ambient temperature or OA loadings. These statements were added in the revised manuscript (Pages 14-15, 337–341).

Table S5. Average relative uncertainties of measurements and calculated parameters.

| Species | $\Delta F/F$[a] | $\Delta A/A$[b] | $\Delta S/S$[c] | $\Delta F\%/F\%$[d] | $\Delta K/K$[e] |
|---|---|---|---|---|---|
| *Isoprene SOA tracers* | | | | | |
| C5-alkene triol 1 | 0.028 | 0.032 | 0.027 | 0.037 | 0.043 |
| C5-alkene triol 2 | 0.028 | 0.054 | 0.033 | 0.036 | 0.059 |
| C5-alkene triol 3 | 0.028 | 0.077 | 0.034 | 0.038 | 0.084 |
| 2-Methylthreitol | 0.028 | 0.051 | 0.028 | 0.033 | 0.059 |
| 2-Methylerythritol | 0.028 | 0.066 | 0.030 | 0.035 | 0.072 |
| *Biomass burning tracer* | | | | | |
| Levoglucosan | 0.051 | 0.16 | 0.054 | 0.072 | 0.17 |
| *Sugars ang sugar alcohols* | | | | | |
| Meso-erythritol | 0.028 | 0.11 | 0.028 | 0.040 | 0.12 |
| Fructose | 0.23 | 0.27 | 0.26 | 0.31 | 0.36 |
| Mannose | 0.045 | 0.27 | 0.049 | 0.062 | 0.28 |
| Glucose | 0.094 | 0.28 | 0.10 | 0.18 | 0.31 |
| Xylitol | 0.10 | 0.12 | 0.10 | 0.14 | 0.16 |
| Arabitol | 0.097 | 0.26 | 0.099 | 0.14 | 0.28 |
| Mannitol | 0.21 | 0.42 | 0.21 | 0.29 | 0.47 |

[a] Particle-phase concentration; [b] gas-phase concentration; [c] total concentration; [d] particle-phase fraction; [e] partitioning coefficients of gas vs. organic and aqueous phases.

**2.** The assumption of LLPS should be discussed. Other studies, for example Pye et al. (2018), could be included in this discussion.

*Response:*

Thanks. We discussed the assumption of LLPS by citing two existing studies (Zuend and Seinfeld, 2012; Pye et al., 2018) in the revised manuscript. Pye et al. (2018) was included in discussions on the agreement between measurement-based and predicted partitioning coefficients.

Page 11, lines 258-262

"Due to the influence of mixing state and water content in aerosols, several studies modeled the gas-particle partitioning of oxygenated organic compounds by defining a liquid-liquid phase separation (LLPS) in the aerosol (Zuend and Seinfeld, 2012; Pye et al., 2018). The organic-inorganic interactions and changes of activity coefficients in aqueous mixtures were fully considered as well."

Pages 19-20, lines 444-464

"When solubility in aerosol liquid water was considered by assuming a LLPS in ambient aerosols, and whenever WSOM and WIOM partitioned into separate (*Case 2*) or single (*Case 3*) liquid phases, the average log $K^{m}_{p,OM}$ of the above mentioned compounds became much closer to or even lay within the range (e.g., levoglucosan) of log $K^{t}_{p,OM}$ (Table 1). These results indicated that the aerosol liquid water (21.3 ± 24.2 µg m$^{-3}$; Table S1) is also an important absorbing phase of ambient polyol tracers in Nanjing. Similarly, the measured average $F\%$ of isoprene SOA tracers in southeastern US and central Amazonia were higher than predictions by assuming instantaneous equilibrium between the gas phase and particulate OM only, and the agreement was improved when parameterization of solubility was included for predictions (Isaacman-

VanWertz et al., 2016). But none of these two studies could reasonably predict the temporal variability of $F$% or log $K^m_{p,OM}$. One possible explanation is that the activity coefficients of isoprene SOA tracers and levoglucosan deviate from unity (0.42–2.04; Pye et al., 2018) and vary with PM composition. Pye et al. (2018) re-analyzed the measurement data from Isaacman-VanWertz et al. (2016) using a thermodynamic equilibrium gas-particle partitioning model in two LLPS modes, which involved organic-inorganic interactions and estimations of activity coefficients as a function of liquid PM mixture composition. The resulting predictions captured both the average and diurnal variations of measured $F$% for polyol tracers, suggesting a necessity in obtaining time-resolved activity coefficients for the implementation of absorptive equilibrium partitioning model."

Page 21, lines 487-490
    "This is in conflict with the fact that the interactions of organic compounds, water, and inorganic ions in aerosols will increase the partitioning of highly oxygenated compounds (O:C$\geq$0.6; e.g., isoprene SOA tracers and levoglucosan) into the particle phase (Pye et al., 2018)."

**3.** I think that the title is somewhat misleading because the answer to the question is actually "neither" for most of the organic markers investigated. I suggest revising the title to reflect this.

*Response:*

    In the revised manuscript, it was shown that the solubility of polyol tracers in aerosol liquid water should not be ignored. Comparisons of measurement-based effective Henry's law constants versus predicted values in pure water indicated increased partitioning toward the particle phase.
    Then the title has been changed into
    "Gas-particle partitioning of polyol tracers at a suburban site in Nanjing, east China: Increased partitioning to the particle phase"

**4.** The comparisons to Denver, CO seem completely random given that meteorology, OA loadings, inorganic composition, and ALWC are quite different between the two locations. I understand that this research group made measurements in both locations, but some additional discussion is warranted to better connect the two locations.

*Response:*

    The observation study in Denver, CO (Xie et al., 2014b) did not measure inorganic composition or estimate ALWC. In that study, evidences showing that gas-particle partitioning of 2-methyltetrol and levoglucosan depended on variations in ambient temperature and absorbing OM mass. Then the influence of aerosol liquid water on partitioning coefficients could not be compared between these two studies. Thus, in the revised manuscript, we only kept the comparisons of measured gas- and particle-phase concentrations. The partitioning coefficients obtained here were discussed by referring to other studies (e.g., Isaacman-VanWertz et al., 2016; Pye et al., 2018)

Pages 19-20, lines 444-464

"When solubility in aerosol liquid water was considered by assuming a LLPS in ambient aerosols, and whenever WSOM and WIOM partitioned into separate (*Case 2*) or single (*Case 3*) liquid phases, the average $\log K^{m}_{p,OM}$ of the above mentioned compounds became much closer to or even lay within the range (e.g., levoglucosan) of $\log K^{t}_{p,OM}$ (Table 1). These results indicated that the aerosol liquid water ($21.3 \pm 24.2$ µg m$^{-3}$; Table S1) is also an important absorbing phase of ambient polyol tracers in Nanjing. Similarly, the measured average $F\%$ of isoprene SOA tracers in southeastern US and central Amazonia were higher than predictions by assuming instantaneous equilibrium between the gas phase and particulate OM only, and the agreement was improved when parameterization of solubility was included for predictions (Isaacman-VanWertz et al., 2016). But none of these two studies could reasonably predict the temporal variability of $F\%$ or $\log K^{m}_{p,OM}$. One possible explanation is that the activity coefficients of isoprene SOA tracers and levoglucosan deviate from unity (0.42–2.04; Pye et al., 2018) and vary with PM composition. Pye et al. (2018) re-analyzed the measurement data from Isaacman-VanWertz et al. (2016) using a thermodynamic equilibrium gas-particle partitioning model in two LLPS modes, which involved organic-inorganic interactions and estimations of activity coefficients as a function of liquid PM mixture composition. The resulting predictions captured both the average and diurnal variations of measured $F\%$ for polyol tracers, suggesting a necessity in obtaining time-resolved activity coefficients for the implementation of absorptive equilibrium partitioning model."

**5.** Several points in the manuscript, including in the 1$^{st}$ sentence of the abstract, the discussion links gas-particle partitioning to source apportionment. However, gas-particle partitioning has importance in the atmosphere that extends way beyond source apportionment (e.g., it affects the PM mass concentration, the lifetime and distribution of organics in the atmosphere, among others). The study thus has broader relevance than is discussed in the manuscript.

*Response:*

Gas/particle partitioning is important in predicting the formation, transport, and life time of organic aerosols in the atmosphere, which are mostly involved in atmospheric transport models (e.g., CMAQ).

Tracer-based source apportionment is a different research method highly dependent on the identification of source-specific tracers. As mentioned in the introduction, the target polyols in this work are firstly known as tracers liked with specific emission sources. But their gas-particle partitioning was poorly characterized. Several measurement studies on gas-particle partitioning of polar organic tracers also emphasized its importance in both modeling and tracer-based source apportionment of organic aerosols (e.g., Zhao et al., 2013; Isaacman-VanWertz et al., 2016). That's the reason gas-particle partitioning of polyol tracers was also linked to source apportionment in some places of the manuscript.

In the abstract, the first sentence has been changed into

"Gas-particle partitioning of water-soluble organic compounds plays a significant role in influencing the formation, transport, and lifetime of organic aerosols in the atmosphere, but is poorly characterized." (Page 2, lines 31-33)

**Technical Corrections**

(1) Line 55: "documented" is not the right word here

*Response:*

Here, "documented for" was replaced by "in". (Page 4, line 78)

(2) Line 65-66: include "GC-MS" in parentheses after introducing the terms

*Response:*

It was revised as suggested. (Page 4, line 89)

(3) Line 80-83: I do not follow the logic of this sentence, please clarify

*Response:*

The original expression has been changed into
"In addition to absorptive partitioning to particulate OM after the formation of oxygenated organic compounds in gas phase, other formation pathways (e.g., reactive uptake) have been identified and are responsible for the large discrepancy between modeled and observed SOA loadings (Jang et al., 2002; Kroll et al., 2005; Perraud et al., 2012)." (Page 5, 104-108)

(4) Line 90: typo in this line

*Response:*

Thanks, "every" should be "very". (Page 5, line 117)

(5) Line 93: suggest changing "depict" to "describe"

*Response:*

It has been changed as suggested. (Page 5, line 120)

(6) Line 101: delete "termed"

*Response:*

It was deleted as suggested.

(7) Line 116: "unveils" is not the right word here

*Response:*

It was replaced by "tends to explain". (Page 6, line 145)

(8) Line 118-120: suggest deleting this sentence

*Response:*

It was deleted as suggested.

(9) Line 134: change "was" to "were"

*Response:*

It was changed as suggested. (Page 7, line 163)

(10) Line 153: change "involving" to "using" or similar

*Response:*

Here, "involving" was replaced by "using". (Page 7, line 179)

(11) Line 221-222: specify that this is theoretical

*Response:*

The whole sentence has been deleted in the revised manuscript.

(12) Line 244: edit sentence for grammar

*Response:*

The original expression has been changed into
"It is therefore suitable to collect gaseous 2-methylterols and levoglucosan using PUF materials only." (Page 13, lines 312-313)

(13) Line 317: edit sentence for grammar

*Response:*

The original expression has been changed into
"………which might be attributed to high levels of vegetation during growing seasons and autumn decomposition (Burshtein et al., 2011)." (Page 17, lines 390-491)

(14) Line 331: delete "data"

*Response:*

It was deleted and the whole paragraph has been rewritten.  (Pages 17-18, lines 397-419)

(15) Line 333: suggest deleting "majorly" and revising the sentence accordingly

*Response:*

The sentence has been changed into

"Gas-phase C5-alkene triols and 2-methyltetrols had maximum concentrations in summer and significant ($p < 0.05$) day-night variations (Figure S4)…………." (Pages 17, lines 402-403)

(16) Line 360: "prediction" should be plural

***Response:***

It was changed as suggested. (Page 19, line 451)

(17) Line 380: change "less stable" to "lower"

***Response:***

We added "lower and" in that sentence. (Page 20, line 478)

(18) Line 386-388: it is quite difficult to follow the discussion here

***Response:***

Here, the comparisons were reorganized and most of the discussions on $K_{p,OM}$ were rewritten.

"When solubility in aerosol liquid water was considered by assuming a LLPS in ambient aerosols, and whenever WSOM and WIOM partitioned into separate (*Case 2*) or single (*Case 3*) liquid phases, the average log $K^m_{p,OM}$ of the above mentioned compounds became much closer to or even lay within the range (e.g., levoglucosan) of log $K^t_{p,OM}$ (Table 1). These results indicated that the aerosol liquid water ($21.3 \pm 24.2$ µg m$^{-3}$; Table S1) is also an important absorbing phase of ambient polyol tracers in Nanjing." (Page 19, lines 444-450)

(19) Line 412: efflorescence RH is more relevant than DRH in this scenario

***Response:***

Here, we deleted the original explanation, and changed the original expression into

"The log ($K^t_{H,w}/K^m_{H,e}$) data increased faster when $c_{sulfate}$ approached 0, and deviated from their expected behavior with increased $c_{sulfate}$. Kampf et al. (2013) selected a threshold $c_{sulfate}$ of 12 mol kg$^{-1}$ ALWC to illustrate the deviation for chamber experiments, and attributed it to elevated viscosity and slow particle-phase reactions at high $c_{sulfate}$." (Pages 22, lines 521-525)

(20) Line 462-463: delete "barely" and revise sentence accordingly

***Response:***

The original expression was changed into

"Then gas-particle partitioning of polyol tracers should have little influence on source apportionment based on particle-phase data in Nanjing." (Page 24, lines 564-566)

(21) Line 465: not sure what exactly is a "concern"? clarification needed

*Response:*

The whole sentence was deleted.

(22) Line 478: delete "pre-"

*Response:*

It was deleted as suggested. (Page 25, line 579)

**References**

Ahrens, L., Shoeib, M., Harner, T., Lane, D. A., Guo, R., and Reiner, E. J.: Comparison of annular diffusion denuder and high volume air samplers for measuring per- and polyfluoroalkyl substances in the atmosphere, Anal. Chem., 83, 9622-9628, 10.1021/ac202414w, 2011.

Isaacman-VanWertz, G., Yee, L. D., Kreisberg, N. M., Wernis, R., Moss, J. A., Hering, S. V., de Sá, S. S., Martin, S. T., Alexander, M. L., Palm, B. B., Hu, W., Campuzano-Jost, P., Day, D. A., Jimenez, J. L., Riva, M., Surratt, J. D., Viegas, J., Manzi, A., Edgerton, E., Baumann, K., Souza, R., Artaxo, P., and Goldstein, A. H.: Ambient gas-particle partitioning of tracers for biogenic oxidation, Environ. Sci. Technol., 50, 9952-9962, 10.1021/acs.est.6b01674, 2016.

Peters, A. J., Lane, D. A., Gundel, L. A., Northcott, G. L., and Jones, K. C.: A comparison of high volume and diffusion denuder samplers for measuring semivolatile organic compounds in the atmosphere, Environ. Sci. Technol., 34, 5001-5006, 10.1021/es000056t, 2000.

Pye, H. O. T., Zuend, A., Fry, J. L., Isaacman-VanWertz, G., Capps, S. L., Appel, K. W., Foroutan, H., Xu, L., Ng, N. L., and Goldstein, A. H.: Coupling of organic and inorganic aerosol systems and the effect on gas–particle partitioning in the southeastern US, Atmos. Chem. Phys., 18, 357-370, 10.5194/acp-18-357-2018, 2018.

Xie, M., Hannigan, M. P., and Barsanti, K. C.: Gas/particle partitioning of n-alkanes, PAHs and oxygenated PAHs in urban Denver, Atmos. Environ., 95, 355-362, http://dx.doi.org/10.1016/j.atmosenv.2014.06.056, 2014a.

Xie, M., Hannigan, M. P., and Barsanti, K. C.: Gas/particle partitioning of 2-methyltetrols and levoglucosan at an urban site in Denver, Environ. Sci. Technol., 48, 2835-2842, 10.1021/es405356n, 2014b.

Zhao, Y., Kreisberg, N. M., Worton, D. R., Isaacman, G., Weber, R. J., Liu, S., Day, D. A., Russell, L. M., Markovic, M. Z., VandenBoer, T. C., Murphy, J. G., Hering, S. V., and Goldstein, A. H.: Insights into secondary organic aerosol formation mechanisms from measured gas/particle partitioning of specific organic tracer compounds, Environ. Sci. Technol., 47, 3781-3787, 10.1021/es304587x, 2013.

Zuend, A., and Seinfeld, J. H.: Modeling the gas-particle partitioning of secondary organic aerosol: the importance of liquid-liquid phase separation, Atmos. Chem. Phys., 12, 3857-3882, 10.5194/acp-12-3857-2012, 2012.

---

## Author Comment (AC3)

This study by Chao Qin et al. reports on filter-based measurements of the gas–particle partitioning of a selection of semi-volatile isoprene oxidation products, levoglucosan and polyols in Nanjing, China. Detailed simultaneous gas and particle phase measurements and assessments of the gas–particle partitioning and influence of aerosol liquid water are relatively scarce. Therefore, this manuscript and the measured data are certainly of interest to the atmospheric chemistry and physics community.

Overall, the manuscript is well written, of adequate lengths and with useful tables and figures. The field sampling and chemical quantification conducted over an extended time span are valuable. The comparison to different predictions by equilibrium gas–particle partitioning models/assumptions is of interest, but it also reveals several issues that need to be addressed.

My main concern is with the provided level of detail on the **uncertainties of the measurements and the theoretical predictions,** as outlined in the general and specific comments below. This manuscript should be (and can be) substantially improved by adding a better discussion and quantification of uncertainties and potential systematic biases as well as clarifications about partitioning mechanisms and involved assumptions. In the present manuscript, the partitioning model discussion is rather confusing, since the title and text suggest a fundamental difference between "absorptive partitioning" and Henry's law partitioning, not recognizing that Henry's law is a way of expressing equilibrium (absorptive) gas–liquid partitioning.

*Response:*
    Thanks for the reviewer's comments, and we will reply these point by point in the reviewers' specific comments.

**General comments**
    The discussion of the presented mismatch between measured and predicted partitioning of several organic tracers in this manuscript would strongly benefit from a more thorough, quantitative uncertainty analysis of the filter measurements and of the assumptions made with the "theoretical" predictions of partitioning coefficients. This would likely lead to relatively wide error bounds on the median and average partitioning coefficients listed in the tables. At present, the study suggests that there is poor agreement with absorptive (Raoult's law) partitioning as well as with solubility-based physical Henry's law partitioning. However, there seem to be substantial uncertainties in the predictions applied and assumptions involved (see the specific comments below).

    A comparison to other studies involving the same or similar compounds should be included. The work by Pye et al. (2018) focuses on measurements and conditions in the southeastern United States and includes field measurements and equilibrium partitioning calculations for several polyols and organic acids in common with this study by Qin et al. Pye et al. (2018) also assessed partitioning of 2-methyltetrol, C5 alkene triol, levoglucosan, pinonic acid and other semivolatile compounds. The Pye et al. work includes predicted or assumed liquid–liquid phase separation cases that differ in phase composition from the assumptions made in this study. Importantly, their results show generally a much better agreement between predicted and measured particle phase fractions. Therefore, it is recommended that the authors compare their findings with those from the Pye et al. study and discuss potential reasons for discrepancies in the

partitioning coefficients and their predictability (or that of particle phase fractions).

*Response:*

In the revised manuscript, we defined three gas-particle partitioning cases. *Case 1* assumed that the particulate OM was the only absorbing material in aerosols based on the equilibrium absorptive partitioning theory. Solubility of polyol tracers in an aqueous phase was included in *Cases 2 and 3*, where water-soluble and water-insoluble OM partitioned into different liquid phases and a single OM phase, respectively. Moreover, measurement uncertainties and their influences on partitioning coefficients of gas versus organic/aqueous phases in aerosols were estimated. It was shown that the average relative uncertainties of measurements and calculated partitioning coefficients ranged from a few percent to ~50%, which corresponded to an uncertainty of less than ± 0.30 for their logarithm values. Although the variability of theoretical partitioning coefficients was large, we still obtained an improved agreement between measurement-based and predicted gas-organic partitioning coefficient ($K_{p,OM}$) for *Cases 2* and *3*. So, aerosol liquid water should have substantial influences on gas-particle partitioning of target polyol tracers in this work.

The interactions of organic and inorganic compounds in the aqueous phase were expected to increase the partitioning of highly water-soluble compounds into the condensed phase (Kroll et al., 2005; Ip et al., 2009; Kampf et al., 2013; Pye et al., 2018). Then the effective Henry's law constants ($K_{H,e}$, mol m$^{-3}$ atm$^{-1}$) of target polyol compounds in aerosols should be greater than those in pure water ($K_{H,w}$). In the revised manuscript, predicted $K_{H,w}$ values from EPI and SPARC estimates varied by several orders of magnitude. Literature $K_{H,w}$ values were closer to those of 2-methyltetrols and levoglucosan estimated by EPI, while the predicted $K_{H,w}$ with SPARC was unreasonably larger than $K_{H,e}$. So, $K_{H,w}$ values based on EPI estimates were used for further data analysis. Because the "salting-in" effect is a known phenomenon that is not likely linked with a specific physical or chemical mechanism, we made some possible explanations (e.g., reactive uptake) for the enhanced uptake of polyol tracers in aerosol liquid water.

Pye et al. (2018) re-evaluated the measurement data of gas- and particle-phase oxygenated compounds in southeastern US using a thermodynamic equilibrium gas-particle partitioning model in two LLPS modes. The modeling work was based on the AIOMFAC model and programed inorganic-organic interactions and variations of activity coefficients as a function of liquid PM mixture composition. The resulting predictions captured both the average and diurnal variability of measured $F\%$ for polyol tracers, suggesting a necessity in obtaining time-resolved activity coefficients for the implementation of absorptive equilibrium partitioning model. In this study, partitioning coefficients of polyol tracers were calculated and predicted empirically assuming equilibrium between gas phase and organic/aqueous phases in aerosols. Moreover, particulate OM phase was presumed as an ideal solution in which the activity coefficient $\zeta_{OM}$ was equal to 1, and no organic-inorganic interactions were considered. So, the temporal variability of $K_{p,OM}$ was poorly predicted in this work, and the gap between $K_{H,e}$ and $K_{H,w}$ could not be explicitly interpreted. Details of the changes were provided in the revised manuscript and responses to specific comments below.

**Specific comments**

**1.** Lines 83 – 87: It is stated that an absorptive partitioning model (which one?) underestimated particle-phase concentrations of carbonyls by several orders of magnitude. Is the argument made by the authors here (from the given phrasing) that absorptive partitioning is an incorrect partitioning mechanism? If so, should it be adsorptive partitioning or what kind? This statement requires further clarification/discussion.

For context, do you mean to say that (1) absorptive partitioning does not take place or (2) that the experiments are not measuring partitioning under equilibrium conditions or that (3) inadequate vapor pressures were used in the partitioning model or (4) something else? For example, could reactive uptake be at play (e.g. mentioned in the cited study by Healy et al., 2008). If the measurement/prediction mismatch is due to reactive uptake, it is questionable to blame absorptive partitioning for this, since that theory may still apply to the parent compound that is partitioning, but further reactions in the particle phase, like hydration of glyoxal, complex formation in presence of sulfate ions or reversible oligomerization may distort the understanding of what species and in what amount is partitioning. It may well be that absorptive equilibrium gas–particle partitioning applies to each of the individual species formed but cannot simply be assumed to be represented by the parent compound considered in the gas phase. Introducing an "effective" Henry's law coefficient can be used to account for the measured partitioning; however, that formulation is then simply a parameterization and not directly elucidating a physical or chemical mechanism.

*Response:*

Thanks. Here we did not intend to state that the absorptive partitioning is an incorrect partitioning mechanism. The mismatch between measurement and prediction could be associated with inappropriate assumptions on absorbing phase (e.g., particulate OM only) and formation pathways (e.g., gas-phase oxidation).

To make it clear, we changed the original expression into

"In addition to absorptive partitioning to particulate OM after the formation of oxygenated organic compounds in gas phase, other formation pathways (e.g., reactive uptake) have been identified and are responsible for the large discrepancy between modeled and observed SOA loadings (Jang et al., 2002; Kroll et al., 2005; Perraud et al., 2012). Unlike non-polar species (e.g., *n*-alkanes, polycyclic aromatic hydrocarbons) and alkanoic acids that are well simulated (Simcik et al., 1998; Xie et al., 2014a; Yatavelli et al., 2014; Isaacman-VanWertz et al., 2016), particle-phase concentrations of carbonyls were underestimated by several orders of magnitude when particulate OM is presumed as the only absorbing phase in ideal condition (Healy et al., 2008; Kampf et al., 2013; Shen et al., 2018)." (Page 5, lines 104-114)

In the original manuscript, the absorptive partitioning means partitioning between gas phase and particulate OM in aerosols; Henry's law partitioning corresponds to the equilibrium between gas phase and aerosol liquid water. We have clarified this throughout the manuscript.

**2.** Line 89: "favored the formation of pinonaldehyde"; do you mean "partitioning" instead of "formation"? The formation of pinonaldehyde (in the gas phase) is likely independent from aerosol water content.

*Response:*

    Zhao et al. (2013) stated that the aerosol water plays a role in the formation of particle-phase pinonaldehyde in the atmosphere. This might be related to the water uptake. So, here we replaced "favored" with "played a role in". (Page 5, lines 116)

**3.** Line 100: The work by Volkamer et al. (2009) on effective Henry's law partitioning and aqueous phase chemistry could also be cited here and perhaps discussed in context of the findings from this study later in the article.

*Response:*

    The work by Volkamer et al. (2009) has been cited here.

    "An effective Henry's law coefficient ($K_{H,e}$, mol m$^{-3}$ atm$^{-1}$) can be used to account for the measured partitioning between the gas phase and aerosol liquid water (Volkamer et al., 2009)." (Page 5-6, lines 124-126).

    Volkamer et al. (2009) focused on the effect of seed chemical composition and photochemistry on SOA yields. They found that the WSOC photochemical reactions can cause increased SOA yield. So, this work was also cited later in the manuscript.

    "Moreover, log ($K^{t}_{H,w}/K^{m}_{H,e}$) values of polyol tracers also negatively correlated with the aqueous-phase concentrations of WSOC ($c_{WSOC}$, Figure S6), but not $NH_4^+$ or $NO_3^-$. This dependence might be associated with the "like-dissolves-like" rule, or indicate the importance of aqueous-phase heterogeneous reactions (Hennigan et al., 2009; Volkamer et al., 2009)." (Pages 23-24, lines 552-556)

**4.** Line 133 – 135: From this description of the gas and aerosol measurements using filters in series, it is not clear how much the uptake of gaseous (semivolatile) organic compounds on accumulated aerosol mass loading of filter 1 (Qf) will contribute to the total concentration on the particle filter. Based on absorptive equilibrium partitioning theory, the accumulated condensed-phase aerosol mass on the first filter may shift the actual gas–particle partitioning in the ambient air to favor additional partitioning from the gas phase to the condensed phase on the filter while the sampling flow passes through the filter, thus possibly leading to a systematic particle phase mass concentration bias. Given the long sampling times, this may constitute a substantial bias. Were such potential issues quantified in controlled experiments? Please discuss.

*Response:*

    Based on absorptive equilibrium partitioning theory, the partitioning coefficient of a certain compound between gas and particulate OM phases ($K_{p,OM}$, m$^3$ ug$^{-1}$) is defined as

$$K_{p,OM} = \frac{F/M_{OM}}{A} \qquad (1)$$

where $F$ and $A$ are (ng m$^{-3}$) are particle- and gas-phase concentrations, and $M_{OM}$ (µg m$^{-3}$) is the mass concentration of particulate OM. It is assumed that the quantity $F/M_{OM}$ (ng µg$^{-1}$) represents the equilibrium concentration in the particulate matter (Pankow and Bidleman, 1992; Liang et al., 1997).

In previous studies, $K_{p,OM}$ values based on offline measurements were typically obtained using sampling periods of many hours (e.g., 8, 12, or 24 h). When ambient concentrations ($F$, $A$, or $M_{OM}$) or temperature change within a sampling interval, the particulate OM initially collected on the filter will tend to re-equilibrate with the $A$ value though evaporation or absorption. Then whether the accumulated aerosol mass will uptake or release gaseous organic compounds depends on how changes in $F$, $A$, $M_{OM}$, and ambient temperature take place. Measured values of $F$, $A$, and $M_{OM}$ will be averages over the whole sampling period, not reflecting real-time atmospheric concentrations. Therefore, the situation raised in the comment seems not applicable to this study.

**5.** Line 152 – 154: "Concentrations of aerosol liquid water were predicted by ISORROPIA II model"; this prediction will only account for water uptake by inorganic ions but neglect any water uptake by hygroscopic organic compounds (such as some WSOC), right? It may therefore lead to an underestimation of the WSOC effect on organic partitioning. The authors could use a simple estimation based on typical organic hygroscopicity parameters (kappa) and the median or actual RH values to estimate the organic-contributed water content by the WSOC mass fraction in particles.

*Response:*

According to Isaacman-VanWertz et al. (2016), the water uptake by WSOC ($W_O$, μg m$^{-3}$) could be estimated as

$$W_O = \frac{V_{WSOC} \times [\kappa \times (O:C)]}{(\frac{100}{RH\%} - 1)} \quad (2)$$

where $V_{WSOC}$ represents WSOC volume, and is calculated as the organic mass (WSOC $\times 1.6$) divided by its density (1.4 g cm$^{-3}$). The hygroscopicity parameter ($\kappa$) and oxygen to carbon ratio (O:C) of WSOC were assumed as 0.10 and 0.5, respectively, based on field and laboratory studies (Taylor et al., 2017; Cai et al., 2020). The resulting $W_O$ had an average of $0.47 \pm 1.14$ μg m$^{-3}$, far below the amount caused by inorganic ions ($21.3 \pm 24.2$ μg m$^{-3}$). Taylor et al. (2017) predicted a growth factor range of 1.00–1.20 with $\kappa$ varying from 60 to ~100%, which lead to a comparable average $W_O$ ($0.42 \pm 0.70$ μg m$^{-3}$) in this work. Thus, the water content contributed by WSOC was not accounted for in this work.

We have clarified this in the revised manuscript and supplementary information. (Page 8, lines 180-182)

"The estimated water content contributed by hygroscopic WSOC was relatively small ($< 1$ μg m$^{-3}$) and not accounted for in this work (Text S1 of supplementary information)."

**6.** Line 209: I suggest adding these equations to the main text.

*Response:*

We have defined three partitioning cases and included these equations in the revised manuscript. (Pages 10 -13, lines 238-295)

"Here, we defined three partitioning cases to explore the influence of dissolution in aerosol liquid water on gas-particle partitioning of polyol tracers in the atmosphere. *Case 1* presumes instantaneous equilibrium between the gas phase and particulate OM

based on the equilibrium absorptive partitioning theory. In this case, particulate OM is assumed to be the only absorbing phase and behave as an ideal solution. Then the absorptive gas-particle partitioning coefficients ($K_{p,OM}$, $m^3$ $\mu g^{-1}$) were calculated from measurements ($K^m_{p,OM}$) and predicted theoretically ($K^t_{p,OM}$) as follows

$$K^m_{p,OM} = \frac{F/M_{OM}}{A} \qquad (2)$$

$$K^t_{p,OM} = \frac{RT}{10^6 \overline{MW}_{OM} \zeta_{OM} p^o_L} \qquad (3)$$

[revised manuscript text omitted]

**7.** Line 212: If I understand your procedure, the ISORROPIA-derived aerosol water content is not accounting for water associated with WSOC, which could be substantial at high RH and when the WSOC represent a significant mass fraction of aerosol. Also, actual interactions among organics and ions within particle phases may affect the partitioning (both between liquid phases and gas/particle), which I assume is ignored in this work. Furthermore, WSOC, while water-extractable by definition, can be of relatively moderate polarity and may preferably partition to the WIOM organic-rich phase in presence of dissolved salts in an aqueous phase (see e.g. Zuend et al., 2012; You et al., 2014; Pye et al. 2018). Hence, it would be useful to estimate errors from such effects on the determined KOW. It may also be adequate to consider other liquid–liquid phase separation scenarios, such as assuming that all WIOM and WSOC organics partitioned to one aqueous organic phase and all inorganic salts to a separate aqueous inorganic phase (compare to Fig. 3 of Pye et al., 2018).

*Response:*

As mentioned in responses to *Comment 5*, the estimated contribution of WSOC to aerosol liquid water is relatively small. In this study, gas-particle partitioning coefficients of polyol tracers were calculated and predicted empirically by assuming equilibrium between gas phase and organic/aqueous phases in aerosols, and the organic-inorganic interactions were not considered. This might be an important reason for the gap between measurement-based $K_{H,e}$ and predicted $K_{H,w}$. Pye et al. (2018) re-analyzed the measurement data from Isaacman-VanWertz et al. (2016) using a thermodynamic equilibrium gas-particle partitioning model in two LLPS modes, which involved inorganic-organic interactions and estimations of activity coefficients as a function of liquid PM mixture composition. The resulting predictions captured both the average and diurnal variations of measured *F*% for polyol tracers, suggesting a necessity in obtaining time-resolved activity coefficients for the implementation of absorptive equilibrium partitioning model. These discussions have been added in the revised manuscript. (Pages 19-20, lines 457-464)

As suggested by the reviewer, we defined a third partitioning case assuming all WIOM and WSOC organics partitioned to a single organic phase (*Case 3*, see responses to *Comment 6*). However, the measurement-based partitioning coefficients of polyol tracers in *Case 3* are very close to those in *Case 2*, where WIOM and WSOC were assumed to partition into separate liquid phases. In the revised manuscript, we defined three gas-particle partitioning cases and re-analyzed the difference between measured and predicted partitioning coefficients. (Pages 10-13, lines 238-295, see responses to *Comment 6*; Sections 3.4 and 3.5 in the revised manuscript, Tables 1 and 2 below).

Table 1. Comparisons of measurement-based log $K_{p,OM}$ (m$^3$ μg$^{-1}$) at three proposed cases and predicted values.

| Species | No. of obs. | Log $K^m_{p,OM}$ [a] | | | Log $K_{p,OM}$ [b] | | | |
|---|---|---|---|---|---|---|---|---|
| | | Case 1 | Case 2 | Case 3 | EPI | EVAPORATION | SPARC | SIMPOL |
| *Isoprene SOA tracers* | | | | | | | | |
| C5-alkene triol 1 | 53 | 0.33 ± 0.71 | -0.79 ± 0.86 | -0.82 ± 0.85 | -3.09 | -2.84 | -1.19 | -2.88 |
| C5-alkene triol 2 | 63 | 0.15 ± 0.55 | -1.02 ± 0.74 | -1.05 ± 0.73 | -3.62 | -3.67 | -4.14 | -2.85 |
| C5-alkene triol 3 | 83 | 0.35 ± 0.68 | -0.83 ± 0.86 | -0.86 ± 0.85 | -2.90 | -2.65 | -1.00 | -2.69 |
| 2-Methylthreitol | 101 | -0.12 ± 0.48 | -2.09 ± 0.71 | -2.09 ± 0.70 | -1.87 | -1.30 | -1.18 | -0.47 |
| 2-Methylerythritol | 95 | -0.011 ± 0.58 | -1.96 ± 0.71 | -1.96 ± 0.71 | -1.90 | -1.34 | -1.22 | -0.50 |
| *Biomass burning tracer* | | | | | | | | |
| Levoglucosan | 65 | 2.23 ± 0.72 | 0.63 ± 0.90 | 0.62 ± 0.90 | -0.04 | -0.81 | 1.04 | -0.76 |
| *Sugars ang sugar alcohols* | | | | | | | | |
| Meso-erythritol | 31 | 0.87 ± 0.53 | -1.43 ± 0.60 | -1.43 ± 0.60 | -0.65 | -1.21 | -0.45 | |
| Fructose | 85 | 0.65 ± 0.73 | -1.20 ± 0.83 | -1.20 ± 0.89 | 1.17 | 2.76 | 6.94 | |
| Mannose | 74 | 0.62 ± 0.71 | -2.12 ± 0.95 | -2.12 ± 0.95 | 1.28 | 2.13 | 4.77 | |
| Glucose | 88 | 0.42 ± 0.67 | -2.77 ± 0.93 | -2.77 ± 0.93 | 0.34 | 3.75 | 7.32 | |
| Xylitol | 22 | 0.24 ± 0.54 | -2.61 ± 0.72 | -2.61 ± 0.72 | 3.37 | 2.34 | 3.57 | |
| Arabitol | 30 | 1.46 ± 0.89 | -1.35 ± 1.24 | -1.35 ± 1.24 | 3.25 | 1.67 | 2.90 | |
| Manitol | 65 | 1.08 ± 0.63 | -2.24 ± 0.95 | -2.24 ± 0.95 | 2.33 | 4.16 | 6.68 | |

[a] Average ± standard deviation; [b] temperature range: -4~36 °C.

Table 2. Comparisons of measurement-based log $K_{H,e}$ (mol m$^{-3}$ atm$^{-1}$) and predicted log $K_{H,w}$ of individual polyol tracers.

| Species | No. of obs. | Log $K^m_{H,e}$ (Cases 2) [a] | | | Log $K^t_{H,w}$ [b] | |
|---|---|---|---|---|---|---|
| | | Median | Average | Range | EPI | SPARC |
| *Isoprene SOA tracers* | | | | | | |
| C5-alkene triol 1 | 53 | 14.0 | 13.9 ± 0.86 | 11.5 – 16.4 | 7.22 | 11.7 |
| C5-alkene triol 2 | 63 | 13.7 | 13.6 ± 0.73 | 11.2 – 16.1 | 7.34 | 7.66 |
| C5-alkene triol 3 | 83 | 13.9 | 13.8 ± 0.85 | 10.6 – 16.1 | 7.43 | 11.9 |
| 2-Methylthreitol | 101 | 13.4 | 13.3 ± 0.70 | 10.9 – 14.8 | 10.0 | 14.1 |
| 2-Methylerythritol | 95 | 13.5 | 13.5 ± 0.71 | 11.6 – 15.6 | 9.95 | 14.1 |
| *Biomass burning tracer* | | | | | | |
| Levoglucosan | 65 | 15.7 | 15.7 ± 0.90 | 13.2 – 17.3 | 13.4 | 16.1 |
| *Sugars ang sugar alcohols* | | | | | | |
| Meso-erythritol | 31 | 14.5 | 14.4 ± 0.60 | 12.8 – 15.6 | 9.65 | 13.8 |
| Fructose | 85 | 14.2 | 14.1 ± 0.89 | 11.9 – 16.5 | 14.7 | 19.9 |
| Mannose | 74 | 14.0 | 14.1 ± 0.94 | 12.1 – 16.8 | 10.9 | 18.8 |
| Glucose | 88 | 13.9 | 13.9 ± 0.93 | 11.3 – 16.3 | 14.7 | 20.9 |
| Xylitol | 22 | 13.8 | 13.7 ± 0.72 | 12.6 – 15.0 | 12.1 | 18.1 |
| Arabitol | 30 | 15.1 | 15.0 ± 1.23 | 13.0 – 18.2 | 11.3 | 17.4 |
| Mannitol | 65 | 14.6 | 14.5 ± 0.94 | 12.1 – 16.4 | 12.9 | 20.8 |

[a] Log $K^m_{H,e}$ values of *Case 3* had ignorable difference, and were not exhibited separately; [b] temperature range: -4~36 °C.

**8.** Lines 221 – 222: The rather low octanol–water partitioning coefficients indicate not only better solubility in water but also that the polyols of moderate to high polarity have low solubility in octanol; this is because octanol is a rather low polarity medium as choice for representing organic aerosol. SOA-rich phases may be of substantially higher polarity than octanol yet still form a separate phase from an aqueous salt-rich phase (e.g. You et al., 2014). This should be acknowledged, and consequences of partitioning assumptions considered in the uncertainty analysis.

*Response:*

As we mentioned in responses to the general comment and specific comment 6, three partitioning cases were proposed in the revised manuscript. In the newly defined *Case 3* where WIOM and WSOC organics partitioned to a single organic phase, the solubility of polyol tracers in the organic phase was expected to increase, and using $K_{OW}$ to calculate the distribution between organic and aqueous phases in aerosols would lead to an underestimation of $K_{p,OM}$, which might not be reasonably adjusted. This was acknowledged in the revised manuscript.

Pages 12-13, lines 288-291.

"Note that the polarity of particulate OM phase in *Case 3* was expected to increase, then using $K_{OW}$ to calculate the distribution of polyols between organic and aqueous phases might lead to underestimated $K^m_{p,OM}$ and overestimated $K^m_{H,e}$."

**9.** Line 224 and SI Eq. (2), Text S2: In the SI, it is stated that for the absorptive partitioning prediction an average organic molar mass MWOMof 200 g/mol was used. This seems to be a common and reasonable assumption, but only for a water-free organic absorbing phase. However, for the partitioning of WSOC compounds when assumed to prefer the aqueous phase, one should account for the low molar mass of water present in substantial amounts in that phase, which would lower the weighted mean molar mass significantly (Liu et al., 2021; Gorkowski et al. 2019). Please consider this and, where applicable, correct the estimated partitioning coefficients.

*Response:*

Among the three proposed partitioning cases in the revised manuscript, *Case 1* assumed that the particulate OM was the only absorbing material in aerosols, ignoring the influence of aerosol liquid water. While in *Cases 2* and *3*, particulate OM and aerosol liquid water were assumed to exist in separate liquid phases, and there was no phase distribution of water. By comparing measurement-based and predicted partitioning coefficients for different cases, we inferred that the aerosol liquid water should play a significant role in influencing gas-particle partitioning of polyol tracers (Table 1, see responses to *Comment 7*). An enhanced uptake of polyol tracers was also identified, which should be closely associated with the organic-inorganic interactions.

In this study, the particulate organic phase was assumed to contain no water, and partitioning of gaseous polyols to organic and aqueous phases in aerosols were assessed separately. So, the influence of water content on molecular weight of particulate organic matter ($MW_{OM}$) was not considered.

**10.** Line 233: Use of the EPI suite estimations should be considered uncertain by about one order of magnitude (or more in certain cases) for predictions involving multifunctional semivolatile compounds. A comparison to other estimation methods for physical Henry's law constants (and their estimated uncertainties) may provide some information on the reliability of this method.

*Response:*

Thanks. We also obtained Henry's law constants of polyols in pure water at 25 ºC

($K^*_{H,w}$) from SPARC (Hilal et al., 2008; http://archemcalc.com/sparc-web/calc) estimates and literatures (Table S4).

Although the $K^*_{H,w}$ (mol m$^{-3}$ atm$^{-1}$) from EPI and SPARC differed by several orders of magnitude, literature values of isoprene SOA and levoglucosan were closer to the estimates of EPI (Table S4). If SPARC $K^*_{H,w}$ values were used, the average log $K^m_{H,e}$ of most polyol tracers would be lower than predictions (log $K^t_{H,w}$, Table 2; see responses to *Comment 7*), indicating that the aqueous phase of ambient aerosol is less hospital to polyol tracers than pure water. This is in conflict with the fact that the interactions of organic compounds, water, and inorganic ions in aerosols will increase the partitioning of highly oxygenated compounds (O:C $\geqslant$ 0.6; e.g., isoprene SOA tracers and levoglucosan) into the particle phase (Pye et al., 2018). Several studies identified a close relationship between salt concentrations of aerosol water and enhanced uptake of very polar compounds (Kroll et al., 2005; Ip et al., 2009; Kampf et al., 2013). Thus, log $K^t_{H,w}$ values of EPI estimates were used for further data analysis. (Pages 20-21, lines 482-494)

**11.** Line 328 – 332: Why was a linear regression/relationship used? Partitioning theory would suggest that it should be a sigmoidal relationship (if applicable), e.g. O'Meara at al (2014); Donahue et al. (2009). However, the partitioning of a specific compound will also depend on the condensed phase absorbing mass concentration (in organic or aqueous phase, as appropriate) and on non-ideality, such as presence of phase separation. Given that only particle phase fraction data above ~80% were determined from the measurements, the expected sigmoidal relationship is perhaps not clear from the data alone.

*Response:*

Thanks. The original Figure 2 (now Figure 3) and associated discussions was changed considering the uncertainties in measurements

[Figure]

Figure 3. Average particle-phase fractions and log $p^{o,*}_L$ of individual polyol tracers. Whiskers represent uncertainties of $F$% and one standard deviation of log $p^{o,*}_L$ derived from different estimation tools.

Pages 17-18, lines 410-419

"In Figure 3, the average $F\%$ uncertainties (6.16–31.2%) of monosaccharides (e.g., fructose) and sugar alcohols (e.g., mannitol) were larger than those of isoprene SOA tracers and levoglucosan (3.33–7.24%) due to their low and variable recoveries (Table S2) and excessive breakthrough (Figure S2). However, the estimated uncertainties of $F\%$ for less volatile polyols ($p^{o,*}_L < \sim 10^{-10}$ atm) were not physically meaningful, as more than 95% of these compounds existed in the particle phase. Considering the uncertainties in $F\%$ and log $p^{o,*}_L$ and high average $F\%$ ($> 85\%$) of target polyol tracers, a dependence of $F\%$ on the vapor pressure could not be determined, and the seasonality and day-night difference ($p > 0.05$) of $F\%$ were obscured."

**12.** Line 334: "their F% values did not show seasonality or day-night difference";
   The range of particle phase fractions observed may not allow for such conclusions if the material is predominantly in the particle phase. Uncertainties in the measurements and temperature dependence of the vapor pressures may mask actual variations.

*Response:*

   The original expression has been changed. See responses to *Comment 11* (Pages 17-18, lines 410-419).

**13.** Lines 339 – 340: "Thus, the changes in vapor pressures with the ambient temperature might not be the main factor driving gas-particle partitioning of polyol tracers in northern Nanjing."
   What about variations in organic aerosol mass concentrations as additional influence?

*Response:*

   Thanks. Xie et al. (2014) found that the gas-particle partitioning of 2-methyltetrols and levoglucosan in urban Denver were dependent on the variations in ambient temperature and absorbing organic matter ($M_{OM}$). So, the original expression was changed into
   "Thus, the changes in vapor pressures with the ambient temperature and/or particulate OM loadings might not be the main factors driving gas-particle partitioning of polyol tracers in Nanjing." (Page 17, lines 408-410)

**14.** Line 360: The re-evaluation of the SV-TAG measurements by Isaacman-VanWertz et al. (2016) in the study by Pye et al. (2018) (see their Fig. 5) involving other models, considerations of vapor pressure adjustments and additional measurement comparisons, shows that higher and lower particle phase fractions were predicted, but that generally the agreement between models and observed F% were consistent across a selection of tracers and much better than the orders of magnitude differences reported in this manuscript.

*Response:*

   When the dissolution of polyols in aerosol water was included, the agreement

between measurement-based and predicted partitioning coefficients of gas vs organic phases ($K_{p,OM}$) were improved substantially. However, the variability of gas-particle partitioning was poorly predicted. The study by Pye et al. (2018) was cited and discussed to explain the discrepancy between measurements and predictions. (Pages 19-20, lines 450-464)

"Similarly, the measured average $F$% of isoprene SOA tracers in southeastern US and central Amazonia were higher than predictions by assuming instantaneous equilibrium between the gas phase and particulate OM only, and the agreement was improved when parameterization of solubility was included for predictions (Isaacman-VanWertz et al., 2016). But none of these two studies could reasonably predict the temporal variability of $F$% or log $K^m_{p,OM}$. One possible explanation is that the activity coefficients of isoprene SOA tracers and levoglucosan deviate from unity (0.42–2.04; Pye et al., 2018) and vary with PM composition. Pye et al. (2018) re-analyzed the measurement data from Isaacman-VanWertz et al. (2016) using a thermodynamic equilibrium gas-particle partitioning model in two LLPS modes, which involved organic-inorganic interactions and estimations of activity coefficients as a function of liquid PM mixture composition. The resulting predictions captured both the average and diurnal variations of measured $F$% for polyol tracers, suggesting a necessity in obtaining time-resolved activity coefficients for the implementation of absorptive equilibrium partitioning model."

**15.** Lines 364 – 367: These statements are misleading and need to be rephrased. Henry's law partitioning is a form of absorptive partitioning (in contrast to adsorptive partitioning). In the case of SVOCs and LVOCs, the difference between vapor–liquid equilibrium and liquid-phase mixing described by using Raoult's law or Henry's law (when accounting for non-ideal mixing) is essentially a matter of choice of reference state (while for non-vapor gases only Henry's law can be applied).

The observed large differences between measurements and different predictions could be the result of a combination of issues and uncertainties associated with the measurements and the models used. If reactive uptake is considered to be the key difference between predictions and measurements, this should be clarified.

*Response:*

Thanks. In the revised manuscript, the absorptive partitioning of gaseous polyols to organic and aqueous phases in aerosols were clearly distinguished, and uncertainties of measurements and predictions and their influences on the calculation of partitioning coefficients were estimated.

Through the comparisons of measurement-based and precited $K_{p,OM}$ before and after the inclusion of polyols dissolution in aerosol water, we inferred that the aerosol liquid water is also an important absorbing phase of ambient polyol tracers in Nanjing. The large gaps of $K^m_{H,e}$ versus $K^t_{H,w}$ could be partly parameterized using the equation defining "salting-in" effects. However, the "salting-in" effect is a known phenomenon that is not likely linked with a specific physical or chemical mechanism. According to existing studies, reactive uptake, aqueous phase reactions, and chemical similarity between partitioning species and the absorbing phase might be responsible for increasing the partitioning of polyol tracers into the condensed phase. In the revised manuscript, we have re-organized and re-written most of the discussions on comparisons between measured and predicted partitioning coefficients. (Sections 3.4

and 3.5, Pages 18-23, lines 420-560).

Here are some discussions on mechanisms related to the gap between $K^{m}_{H,e}$ and $K^{t}_{H,w}$. (Pages 23-24, lines 534-560)

"However, the "salting-in" effect is a known phenomenon that is not likely linked with a specific physical or chemical mechanism. Quantum chemical calculation results indicated negative Gibbs free energy of water displacement for interactions between $SO_4^{2-}$ and glyoxal monohydrate (Waxman et al., 2015). The net "salting-in" effect of 1-nitro-2-naphthol in NaF solution was interpreted by postulating hydrogen bonding (Almeida et al., 1983). A direct binding of cations to ether oxygens was proposed to be responsible for the increased solubility of water-soluble polymers (Sadeghi and Jahani, 2012). Due to the moderate correlations and negative intercepts in Figures 4 and S5, the gap between $K^{t}_{H,e}$ and $K^{m}_{H,w}$ cannot be closed by the "salting-in" effect alone. Shen et al. (2018) also obtained negative intercepts when plotting log ($K^{t}_{H,w}/K^{m}_{H,e}$) over $c_{sulfate}$ for glyoxal and methylglyoxal in ambient atmosphere, and attributed this to unknown gas-particle partitioning mechanisms. Evidences showing that the thermal degradation of less volatile oligomers and organosulfates can lead to an overestimation of 2-methyltetrols by 60–188% when using a conventional GC/EI-MS method (Cui et al., 2018). To fit the gas-particle distribution of 2-methyltetrols in southeastern US, 50% of particulate 2-methyltetrols was presumed to exist in chemical forms with much lower vapor pressures by Pye et al. (2018). So, the reactive uptake and aqueous phase chemistry could be explanations for the enhanced uptake of isoprene SOA tracers. Moreover, log ($K^{t}_{H,w}/K^{m}_{H,e}$) values of polyol tracers also negatively correlated with the aqueous-phase concentrations of WSOC ($c_{WSOC}$, Figure S6), but not $NH_4^+$ or $NO_3^-$. This dependence might be associated with the "like-dissolves-like" rule, or indicate the importance of aqueous-phase heterogeneous reactions (Hennigan et al., 2009; Volkamer et al., 2009). Although several studies have estimated Henry's law constants for a variety of polar organic compounds in pure water (e.g., polyols and polyacids; Compernolle and Müller, 2014a, b), more work is warranted to decrease the estimation uncertainty and explain their increased partitioning toward aerosol liquid water explicitly."

**16.** Lines 387 – 389: The statements on these lines seem to support the conclusion that absorptive partitioning may be applicable to describing the partitioning of these isoprene SOA tracers, but only if one uses the "appropriate" absorbing organic phase mass in the estimation of the measured Kp values (and given the uncertainty in the vapor pressures and activity coefficients, this seems to be reasonable). The phrasing could be improved to make that point.

*Response:*

Yes, the agreement between measurement-based and predicted $K_{p,OM}$ will be significantly improved when the "appropriate" absorbing organic phase is used.

To make it clear, we defined three partitioning cases in the revised manuscript. *Case 1* assumed that the particulate OM was the only absorbing material in aerosols based on the equilibrium absorptive partitioning theory. Solubility of polyol tracers in an aqueous phase was included in *Cases 2 and 3*, where water-soluble and water-insoluble OM partitioned into different liquid phases and a single OM phase, respectively. We found a much better agreement between measurement-based and precited $K_{p,OM}$ after considering polyols dissolution in aerosol liquid water, indicating

that aerosol liquid water is also an important absorbing phase of ambient polyols in Nanjing.

Pages 18-19, lines 421-450

"To understand if particulate OM is the only absorbing phase in aerosols for polyol tracers in Nanjing, the absorptive partitioning coefficients of gas vs. organic phases were calculated based on measurement results ($K^m_{p,OM}$) for predefined *Cases 1-3* and predicted theoretically ($K^t_{p,OM}$) using eq. 3 and vapor pressures listed in Table S4. In Table 1, $K^t_{p,OM}$ ranges of isoprene SOA tracers, levoglucosan, and meso-erythritol are within two orders of magnitude, while those of monosaccharides and mannitol are larger ($> 10^3$).When particulate OM was assumed as the only absorbing phase (*Case 1*), the average $K^m_{p,OM}$ of isoprene SOA tracers, levolgucosan, and meso-erythritol were more than 10 times greater than most of their $K^t_{p,OM}$ (Table 1), and this difference was not likely susceptible to measurement uncertainties. As shown in Table S5, the average relative uncertainties of measurement-based partitioning coefficients are all <50%, leading to an uncertainty of log $K^m_{p,OM}$ less than ± 0.30.. …………………………………………

When solubility in aerosol liquid water was considered by defining a LLPS in ambient aerosols, and whenever WSOM and WIOM partitioned into separate (*Case 2*) or single (*Case 3*) phases, the average log $K^m_{p,OM}$ of the above mentioned compounds became much closer to or even lay within the range (levoglucosan) of log $K^t_{p,OM}$ (Table 1). These results indicated that the aerosol liquid water (21.3 ± 24.2 µg m$^{-3}$; Table S1) is also an important absorbing phase of ambient polyol tracers in Nanjing."

**17.** Lines 415 – 421: The finding that the intercept in Fig. 3 of the linear regression does not go through 0.0 indicates that there are substantial uncertainties, making this comparison far less convincing. The scatter in the data is large, also hinting at salting-in as an effect alone does not seem to be a good explanation of the deviations between predicted and measurement-derived Henry's law partitioning. The authors also mention this on lines 445 – 448. There may be other confounding factors that happen to correlate with sulfate concentration; leading to a spurious conclusion of a causal salting-in effect that is not strongly supported by the provided evidence. For example, the ratio of WIOC to WSOC organic material may correlate with sulfate concentrations since sulfate and ammonia amounts will affect and respond to aerosol pH, which may also correlate with RH and absolute ALWC (Pye et al., 2020). Did the authors consider this?

Furthermore, a salting-in of polyols by sulfate is a finding that would be contrary to other studies on liquid–liquid phase partitioning involving polyols and ammonium sulfate, e.g. see Table 1 of Marcolli and Krieger (2006). In the study by Marcolli and Krieger (2006), ammonium sulfate led to salting-out while ammonium nitrate was found to show a salting-in effect on polyols. However, the complexity of the samples from Nanjing, where perhaps acidity and other aerosol components affect uptake, may differ from those in laboratory experiments by Marcolli and Krieger. Please discuss your findings of potential reasons for the model–measurement discrepancies and sulfate influence also in comparison to findings on salting-in/out from those studies.

*Response:*

Although the "salting-in" effect has been known for a long time, it is poorly characterized at high salt concentrations and is not understood mechanistically. The

"salting-in" effect could be considered as a phenomenon that is not specifically linked with a physical or chemical mechanism. So, the equation defining "salting-in" effects was only used to parameterize the enhanced uptake of polyols in aerosol liquid water. In the revised manuscript, we provided some guesses on mechanisms related to the "salting-in" effect from previous work and other explanations for increased partitioning to the particle phase.

Pages 23-24, lines 534-560

"However, the "salting-in" effect is a known phenomenon that is not likely linked with a specific physical or chemical mechanism. Quantum chemical calculation results indicated negative Gibbs free energy of water displacement for interactions between $SO_4^{2-}$ and glyoxal monohydrate (Waxman et al., 2015). The net "salting-in" effect of 1-nitro-2-naphthol in NaF solution was interpreted by postulating hydrogen bonding (Almeida et al., 1983). A direct binding of cations to ether oxygens was proposed to be responsible for the increased solubility of water-soluble polymers (Sadeghi and Jahani, 2012). Due to the moderate correlations and negative intercepts in Figures 4 and S5, the gap between $K^t_{H,e}$ and $K^m_{H,w}$ cannot be closed by the "salting-in" effect alone. Shen et al. (2018) also obtained negative intercepts when plotting log ($K^t_{H,w}/K^m_{H,e}$) over $c_{sulfate}$ for glyoxal and methylglyoxal in ambient atmosphere, and attributed this to unknown gas-particle partitioning mechanisms. Evidences showing that the thermal degradation of less volatile oligomers and organosulfates can lead to an overestimation of 2-methyltetrols by 60–188% when using a conventional GC/EI-MS method (Cui et al., 2018). To fit the gas-particle distribution of 2-methyltetrols in southeastern US, 50% of particulate 2-methyltetrols was presumed to exist in chemical forms with much lower vapor pressures by Pye et al. (2018). So, the reactive uptake and aqueous phase chemistry could be explanations for the enhanced uptake of isoprene SOA tracers. Moreover, log ($K^t_{H,w}/K^m_{H,e}$) values of polyol tracers also negatively correlated with the aqueous-phase concentrations of WSOC ($c_{WSOC}$, Figure S6), but not $NH_4^+$ or $NO_3^-$. This dependence might be associated with the "like-dissolves-like" rule, or indicate the importance of aqueous-phase heterogeneous reactions (Hennigan et al., 2009; Volkamer et al., 2009). Although several studies have estimated Henry's law constants for a variety of polar organic compounds in pure water (e.g., polyols and polyacids; Compernolle and Müller, 2014a, b), more work is warranted to decrease the estimation uncertainty and explain their increased partitioning toward aerosol liquid water explicitly."

As shown in the Figure below, concentrations of sulfate do not correlate with WIOC/WSOC ratios, RH, or ALWC in this work, so the condition raised in the comment was not considered.

[Figure]

Marcolli and Krieger (2006) found that ammonium sulfate (AS) and NaCl are "salting-out" agent for alcohols with medium hydrophilicity, including glycerol, 1,4-butanediol, 1,2-hexanediol, and PEG400. But all the target polyols in this study are

highly water soluble, and there is no evidence showing that AS is salting out agent for any of the polyols studied in this work. A number of studies identified a close relationship between salt concentrations of aerosol water and enhanced uptake of very polar compounds (Setschenow 1889; Almeida et al., 1983; Kroll et al., 2005; Ip et al., 2009; Kampf et al., 2013; Shen et al., 2018). The particle-phase interactions of organic compounds, water, and inorganic ions were found to increase partitioning of isoprene SOA tracers and levoglucosan into the condensed phase (Pye et al., 2018). Since none of the polyol compounds in Marcolli and Krieger (2006) was studied in this work, the moderate dependence of increased uptake on sulfate concentrations is not contrary to their findings, and we did not compare the study results between these two studies.

In the revised manuscript, we re-organized section 3.5 (*Partitioning coefficients of gas versus aqueous phases*) and rewrote most part of it.

**18.** Lines 445 – 448: related to the previous comment, here the authors state that the large gap between KH,e and KH,w cannot be explained by salting-in by sulfate alone. This confirms my impression that the discussion about reasons of the substantial deviations is rather speculative. The presented analyses do not support a firm conclusion about absorptive or non-absorptive partitioning. Moreover, if the effective Henry's law coefficient obtained is due to reactive uptake and/or aqueous phase chemistry, such as oligomer formation, then enhanced particle-phase fractions would be a reasonable expectation. However, a key question would then be whether such chemistry would be reversible during the quantification of the filter material, such that an oligomerized species would appear as monomers, since otherwise it should not contribute to the parent species' particle phase amount. This reviewer would appreciate some discussion about this.

*Response:*

Reactive uptake is very likely contributing to increased partitioning of polyols into the particle phase. In the revised manuscript, we have clarified that the "salting-in" effect is a is a known phenomenon that is not specifically linked with a physical or chemical mechanism (Pages 23-24, lines 534-560, See responses to *Comments 15 and 17*).

The equation defining "salting-in" effects was only used to parameterize the enhanced uptake of polyols in aerosol liquid water (Pages 22-23, lines 517-533).

"As sulfate has been identified as the major factor influencing the salting effect of carbonyl species (Kroll et al., 2005; Ip et al., 2009), Figure 4 shows modified Setschenow plots for C5-alkene triols, 2-methyltetrols, and levoglucosan, where log $(K^t_{H,w}/K^m_{H,e})$ values were regressed to the molality of sulfate ion in aerosol liquid water ($c_{sulfate}$, mol kg$^{-1}$ ALWC). The log $(K^t_{H,w}/K^m_{H,e})$ data increased faster when $c_{sulfate}$ approached 0, and deviated from their expected behavior with increased $c_{sulfate}$. Kampf et al. (2013) selected a threshold $c_{sulfate}$ of 12 mol kg$^{-1}$ ALWC to illustrate the deviation for chamber experiments, and attributed it to elevated viscosity and slow particle-phase reactions at high $c_{sulfate}$. In Figure 4, negative correlations ($p < 0.01$) are observed at $c_{sulfate} < 12$ mol kg$^{-1}$ ALWC, and Figure S5 exhibits significant negative correlations ($p < 0.01$) between log $(K^t_{H,w}/K^m_{H,e})$ and $c_{sulfate}$ for individual polyols even without excluding the deviations at high $c_{sulfate}$. The $K_s$ values of polyol tracers from Figures 4 and S5 (-0.17 – -0.037 kg mol$^{-1}$) are in a similar range as that of glyoxal (-0.24 – -0.04 kg mol$^{-1}$; Kampf et al., 2013; Shen et al., 2018; Waxman et al., 2015). These results

indicated that the shifting of gas-particle equilibrium toward the condensed phase might be partly parameterized by the equation defining "salting-in" effects."

Moreover, we added some discussions about the influence of reactive uptake/aqueous chemistry on enhanced particle-phase concentrations. (Pages 23-24, lines 545-556)

"Evidences showing that the thermal degradation of less volatile oligomers and organosulfates can lead to an overestimation of 2-methyltetrols by 60–188% when using a conventional GC/EI-MS method (Cui et al., 2018). To fit the gas-particle distribution of 2-methyltetrols in southeastern US, 50% of particulate 2-methyltetrols was presumed to exist in chemical forms with much lower vapor pressures by Pye et al. (2018). So, the reactive uptake and aqueous phase chemistry could be explanations for the enhanced uptake of isoprene SOA tracers. Moreover, log ($K^t_{H,w}/K^m_{H,e}$) values of polyol tracers also negatively correlated with the aqueous-phase concentrations of WSOC ($c_{WSOC}$, Figure S6), but not $NH_4^+$ or $NO_3^-$. This dependence might be associated with the "like-dissolves-like" rule, or indicate the importance of aqueous-phase heterogeneous reactions (Hennigan et al., 2009; Volkamer et al., 2009)."

**19.** Table 1. Units of $K_{p,OM}$ should be provided; also state the temperature range for the values shown. Same for Table 2 and other such table in the Supplementary Information document.

*Response:*

Unites and temperatures were added in all related Tables as suggested (See responses to *Comment 7*).

**20.** Figure 2: Assuming a form of absorptive vapor–liquid equilibrium partitioning, the fraction in the particle phase of a semi-volatile organic will not only depend on the pure component vapor pressure but also on the aerosol mass concentration of the absorbing phase (and its composition). Therefore, it would make sense to state the aerosol mass concentration range that was used from the measurements. This would also allow for better comparison to other field measurements.

*Response:*

In this work, mass concentrations of OC were directly obtained from measurements, and were used for the calculation of $K_{p,OM}$. The concentration range of OC (2.24 – 16.8 $\mu g\ m^{-3}$) was added as suggested (now Figure 3, See responses to *Comment 11*).

**Supplementary Information (SI):**

**21.** Text S2: activity coefficients were assumed to be unity for all species in each sample. Is that a justified assumption? Consider that activity coefficients could be far from unity for compounds that are moderately polar (between the polarity of water and that of hydrophobic organics) used for characterizing the two particle phases in this work. This might contribute an order of magnitude of uncertainty for some compounds, but little

for others.

*Response:*

In this work, the organic phase in aerosols was assumed to behave as ideal solution, and the variability of activity coefficient as a function of PM composition was not considered. This should be one reason the variations of measurement-based $K_{p,OM}$ were poorly characterized in this work. Mean activity coefficients of isoprene SOA tracers and levoglucosan in organic-rich phases had a range of 0.42 to 2.04 based on AIOMFAC predictions (Pye et al., 2018), then assuming the activity coefficient to be unity will contribute an uncertainty far less than an order of magnitude.

We added some discussions to state the influence of variability in activity coefficients in the revised manuscript.

Pages 19-20, lines 450-464.

"Similarly, the measured average $F$% of isoprene SOA tracers in southeastern US and central Amazonia were higher than predictions by assuming instantaneous equilibrium between the gas phase and particulate OM only, and the agreement was improved when parameterization of solubility was included for predictions (Isaacman-VanWertz et al., 2016). But none of these two studies could reasonably predict the temporal variability of $F$% or log $K^m_{p,OM}$. One possible explanation is that the activity coefficients of isoprene SOA tracers and levoglucosan deviate from unity (0.42–2.04; Pye et al., 2018) and vary with PM composition. Pye et al. (2018) re-analyzed the measurement data from Isaacman-VanWertz et al. (2016) using a thermodynamic equilibrium gas-particle partitioning model in two LLPS modes, which involved organic-inorganic interactions and estimations of activity coefficients as a function of liquid PM mixture composition. The resulting predictions captured both the average and diurnal variations of measured $F$% for polyol tracers, suggesting a necessity in obtaining time-resolved activity coefficients for the implementation of absorptive equilibrium partitioning model."

**22.** Text S2, below Eq. (3): why is it the "subcooled" liquid vapor pressure? It would be sufficient to denote it as the liquid vapor pressure or liquid-state vapour pressure. Whether it is subcooled/supercooled at given temperature or just a "regular" liquid state does not matter.

*Response:*

The term "subcooled liquid vapor pressure" has been changed into "liquid-state vapor pressure".

**23.** Also, given the relatively large uncertainty associated with vapor pressure estimation methods (O'Meara et al., 2014), it may be advised to compare those values to predictions from other methods (e.g. using the UManSysProp online tools). Uncertainties in pure-component vapor pressures could contribute more than one order of magnitude of uncertainty to Kp estimates.

*Response:*

Thanks. Pure-compound vapor pressures were estimated using a variety estimation tools (EPI, Evaporation, SPARC, and SIMPOL; Table S4), and the resulting $K_{p,OM}$ predictions were compared with measurement-based values in Table 1 of the revised manuscript (See responses to *Comments 7*). Descriptions of the results and associated discussions were rewritten (Section 3.4).

Pages 18-19, lines 420-464

"To understand if particulate OM is the only absorbing phase in aerosols for polyol tracers in Nanjing, the absorptive partitioning coefficients of gas vs. organic phases were calculated based on measurement results ($K^m_{p,OM}$) for predefined *Cases 1-3* and predicted theoretically ($K^t_{p,OM}$) using eq. 3 and vapor pressures listed in Table S4. In Table 1, $K^t_{p,OM}$ ranges of isoprene SOA tracers, levoglucosan, and meso-erythritol are within two orders of magnitude, while those of monosaccharides and mannitol are larger ($> 10^3$). When particulate OM was assumed as the only absorbing phase (*Case 1*), the average $K^m_{p,OM}$ of isoprene SOA tracers, levolgucosan, and meso-erythritol were more than 10 times greater than most of their $K^t_{p,OM}$ (Table 1), and this difference was not likely susceptible to measurement uncertainties. As shown in Table S5, the average relative uncertainties of measurement-based partitioning coefficients are all <50%, leading to an uncertainty of log $K^m_{p,OM}$ less than $\pm$ 0.30. Comparable or even greater (up to $10^5$) gap between $K^m_{p,OM}$ and $K^t_{p,OM}$ has been observed for carbonyls in a number of laboratory and field studies (Healy et al., 2008; Zhao et al., 2013; Shen et al., 2018), which could be ascribed to reactive uptake (e.g., hydration, oligomerization, and esterification) of organic gases onto condensed phase (Galloway et al., 2009). Oligomers, sulfate and nitrate esters of 2-methyltetrols can be formed in the aerosol phase (Surratt et al., 2010; Lin et al., 2014), and their decomposition and hydrolysis during filter analysis will lead to an overestimation of particle-phase concentrations (Lin et al., 2013; Cui et al., 2018). However, the occurrence of oligomers, sulfate or nitrate esters of levoglucosan was not ever reported in ambient aerosols, although it can be readily oxidized by •OH in the aqueous phase of atmospheric particles (Hennigan et al., 2010; Hoffmann et al., 2010).

When solubility in aerosol liquid water was considered by assuming a LLPS in ambient aerosols, and whenever WSOM and WIOM partitioned into separate (*Case 2*) or single (*Case 3*) liquid phases, the average log $K^m_{p,OM}$ of the above mentioned compounds became much closer to or even lay within the range (e.g., levoglucosan) of log $K^t_{p,OM}$ (Table 1). These results indicated that the aerosol liquid water (21.3 $\pm$ 24.2 µg m$^{-3}$; Table S1) is also an important absorbing phase of ambient polyol tracers in Nanjing. Similarly, the measured average $F$% of isoprene SOA tracers in southeastern US and central Amazonia were higher than predictions by assuming instantaneous equilibrium between the gas phase and particulate OM only, and the agreement was improved when parameterization of solubility was included for predictions (Isaacman-VanWertz et al., 2016). But none of these two studies could reasonably predict the temporal variability of $F$% or log $K^m_{p,OM}$. One possible explanation is that the activity coefficients of isoprene SOA tracers and levoglucosan deviate from unity (0.42–2.04; Pye et al., 2018) and vary with PM composition. Pye et al. (2018) re-analyzed the measurement data from Isaacman-VanWertz et al. (2016) using a thermodynamic equilibrium gas-particle partitioning model in two LLPS modes, which involved organic-inorganic interactions and estimations of activity coefficients as a function of liquid PM mixture composition. The resulting predictions captured both the average and diurnal variations of measured $F$% for polyol tracers, suggesting a necessity in obtaining time-resolved activity coefficients for the implementation of absorptive equilibrium partitioning model."

**24.** Figure S5: Please state the vapor pressure units and reference temperature used for the stated Log (p0L) values (one should not have to go back to the text to search for these).

*Response:*

    The vapor pressure unit (atm) and reference temperature (25$^{\circ}$C) were added in the figure caption of Figure S5 (now Figure S4).

    "Figure S4. Temporal variations of gas-phase concentrations and particle-phase fractions ($F\%$) of polyol tracers. $p^{\text{o,*}}_{\text{L}}$: Liquid-state vapor pressure (atm, EPI estimates) at 25 $^{\circ}$C."

---

## Author Response (AR2)

**Reviewer 1**

I thank the authors for their more detailed examination of error and uncertainty, as well as their more careful consideration of uncertainty in theoretical partitioning values. These revisions significantly improve the manuscript, and the authors have reasonably tempered some of their original claims based on these new insights.

I highlight a few specific remaining concerns below, the first of which absolutely needs to be addressed prior to publication while the others are suggestions to add clarity.

**1.** Equation S1 used by the authors to calculate "organic water" is not correct as written. Instead of kappa*O:C, the equation should estimate kappa from O:C, e.g., as 0.29*O:C, based on the work by Massoli et al. correlating kappa to O:C (https://doi.org/10.1029/2010GL045258). This change will roughly triple the calculated W_o, which will not significantly alter the conclusion that organic water is a minor component, but it needs to be corrected before publication.

*Response:*

Thanks. Here Eq. S1 was corrected into

$$W_O = \frac{V_{WSOC} \times \kappa}{(\frac{100}{RH\%} - 1)} \qquad (1)$$

where $V_{WSOC}$ represents WSOC volume, and is calculated as the organic mass (WSOC ×1.6) divided by its density (1.4 g cm⁻³). The hygroscopicity parameter ($\kappa$) of secondary organic aerosol (SOA) was observed to increase as a function of its oxygen to carbon ratio (O:C; Massoli et al., 2010). Isaacman-VanWertz et al. (2016) estimated $\kappa$ of hygroscopic organic matter by assuming a linear regression slope of 0.29 between $\kappa$ and O:C. In previous field and laboratory studies, $\kappa$ values of hygroscopic organic matter typically varied from 0.05 to 0.25 with a O:C range of 0.3–1.0 (Chang et al., 2010; Massoli et al., 2010; Taylor et al., 2017), and the corresponding average $W_O$ based on Eq. S1 in this work ranged from 0.47 ± 1.14 to 2.34 ± 4.03 µg m⁻³, far below the amount caused by inorganic ions (21.3 ± 24.2 µg m⁻³, Table S1). Taylor et al. (2017) predicted a hygroscopic growth factor range of 1.00–1.20 with RH varying from 60 to ~100%, which lead to an average $W_O$ of 0.42 ± 0.70 µg m⁻³ for this study. Thus, the water content contributed by WSOC was not accounted for in this work.

In Text S1 of supplementary information, the original expression was changed accordingly.

**2.** It's not clear to me why the authors use "S", "F", and "A" for total, particle, and gas concentrations, respectively, in units of mass per volume air. They also use M_OM for total organic mass per volume air. All of these are mass concentrations, and it seems to me they should use unified notation (e.g., M_i,tot, M_i,p , and M_i,g ). In the end, that is a style choice, but as written does not follow typical chemistry notation guidance (e.g., IUPAC).

*Response:*

In previous studies where the gas-particle partitioning coefficient ($K_p$) was defined and calculated (e.g., Pankow, 1994a, b; Liang and Pankow, 1996; Liang et al., 1997; Mader and Pankow, 2002), "*F*" and "*A*" were always used to represent particle- and gas-phase concentrations, respectively, in units of mass per volume air. "*S*" was used to denote the sum of *F* and *A* in our previous work (Xie et al., 2013, 2014), where the influence of gas-particle partitioning on receptor-based source apportionment of organic aerosols was evaluated. To keep consistent with existing studies, "*S*", "*F*", and "*A*" were used for total, particle, and gas concentrations, respectively. Since all abbreviations have been clearly defined, the usage of "*S*", "*F*", and "*A*" will not introduce any ambiguity.

In the revised manuscript and supplementary information, we added some references for the definition of $K_p$ (Page 11, lines 245-246) and "*S*" (or "Δ*S*"; Text S5 in supplementary information)

**3.** Line 545-548 has odd grammar, should be corrected.

*Response:*

Thanks, the original expression was changed into

"There is evidence showing that conventional GC/EI-MS analysis overestimates particle-phase 2-methyltetrols by 60-188% due (in part) to the thermal degradation of less volatile oligomers and organosulfates (Cui et al., 2018)". (Page 23, lines 546-549)

**Reviewer 2**

The authors have responded to the comments by referees and have made substantial changes and additions, which have clarified and improved this manuscript. The revisions are well done.

I appreciate the extra work put into quantifying uncertainties and the evaluation of gas-particle partitioning data based on three different cases. I have only a few minor comments below. I recommend this manuscript for publication in ACP.

Specific comments:

**1)** In the response/changed text the authors write "… particle-phase concentrations of carbonyls were underestimated by several orders of magnitude when particulate OM is presumed as the only absorbing phase in ideal condition (Healy et al., 2008; Kampf et al., 2013; Shen et al., 2018)."

- Perhaps rephrase "in ideal condition" to "assuming ideal mixing conditions"

*Response:*

Thanks, the original expression was changed into

"…..particle-phase concentrations of carbonyls were underestimated by several orders of magnitude when particulate OM is presumed as the only absorbing phase assuming ideal mixing condition (Healy et al., 2008; Kampf et al., 2013; Shen et al., 2018)." (Page 5, line 113)

**7)** Tables 1,2: Good to have these comparisons in the manuscript. Make sure to indicate whether Log is base-10 log or natural log.

*Response:*

We stated that the logarithm values were using a base of 10 in the footnotes of Tables 1 and 2.

**17)** I consider glycerol to be among the group of highly water-soluble compounds (it is miscible in all proportions with water, i.e. infinitely soluble); therefore, it may be similar to the solubility of some of the "highly soluble" polyols studied in the present work. I agree that the weak correlation of polyol uptake with sulfate may not be in contradiction to the work by Marcolli and Krieger (2006), but only if salting-in by ammonium sulfate is not proposed to be the dominant mechanism (otherwise it would seem contradictory).

*Response:*

Although the ammonium sulfate (AS) is a "salting-out" agent for glycerol in water solution, it might not be appropriate to infer that AS is a "salting-out" agent for other water-soluble polyols in aerosol liquid water. The interactions of inorganic ions and organic compounds in the aqueous phase of aerosols might increase the partitioning of highly oxygenated compounds into the particle phase through reactive uptake and aqueous phase chemistry, but not the changes in water-solubility. That's why we consider the "salting-in" effect to be a phenomenon, but not a specific physical or chemical mechanism in this version of the manuscript (Page 23, lines 535-536).

"However, the "salting-in" effect is a known phenomenon that is not likely linked with a specific physical or chemical mechanism."

The definition equation for "salting-in" effect was only used to parameterize the increased partitioning of gas-phase polyols to the aqueous phase in aerosols (Page 23, lines 532-534), of which the potential underlying mechanisms were discussed separately (Pages 23-24, lines 546-557).

"There is evidence showing that conventional GC/EI-MS analysis overestimates particle-phase 2-methyltetrols by 60-188% due (in part) to the thermal degradation of less volatile oligomers and organosulfates (Cui et al., 2018). To fit the gas-particle distribution of 2-methyltetrols in southeastern US, 50% of particulate 2-methyltetrols was presumed to exist in chemical forms with much lower vapor pressures by Pye et al. (2018). So, the reactive uptake and aqueous phase chemistry could be explanations for the enhanced uptake of isoprene SOA tracers. Moreover, log ($K^t_{H,w}/K^m_{H,e}$) values of polyol tracers also negatively correlated with the aqueous-phase concentrations of WSOC ($c_{WSOC}$, Figure S6), but not $NH_4^+$ or $NO_3^-$. This dependence might be associated with the "like-dissolves-like" rule, or indicate the importance of aqueous-phase heterogeneous reactions (Hennigan et al., 2009; Volkamer et al., 2009)."

In both the abstract and conclusions, we stated that the enhanced uptake of polyol tracers in aerosol liquid water might be ascribed to mechanisms of reactive uptake, aqueous phase reaction, "like-dissolves-like" principle, etc. (Pages 2-3, lines 52-55; Page 24, lines 572-574) "Salting-in" by ammonium sulfate was not proposed to be the dominant mechanism for increasing the partitioning of polyol tracers into the condensed phase.

[revised manuscript text omitted]